# Orderly assembly underpinning built-in asymmetry in the yeast centrosome duplication cycle requires cyclin-dependent kinase

Marco Geymonat[1†], Qiuran Peng[1†], Zhiang Guo[1], Zulin Yu[2], Jay R Unruh[2], Sue L Jaspersen[2,3]*, Marisa Segal[1]*

[1]Department of Genetics, University of Cambridge, Cambridge, United Kingdom; [2]Stowers Institute for Medical Research, Kansas City, United States; [3]Department of Molecular and Integrative Physiology, University of Kansas Medical Center, Kansas City, United States

**Abstract** Asymmetric astral microtubule organization drives the polarized orientation of the *S. cerevisiae* mitotic spindle and primes the invariant inheritance of the old spindle pole body (SPB, the yeast centrosome) by the bud. This model has anticipated analogous centrosome asymmetries featured in self-renewing stem cell divisions. We previously implicated Spc72, the cytoplasmic receptor for the gamma-tubulin nucleation complex, as the most upstream determinant linking SPB age, functional asymmetry and fate. Here we used structured illumination microscopy and biochemical analysis to explore the asymmetric landscape of nucleation sites inherently built into the spindle pathway and under the control of cyclin-dependent kinase (CDK). We show that CDK enforces Spc72 asymmetric docking by phosphorylating Nud1/centriolin. Furthermore, CDK-imposed order in the construction of the new SPB promotes the correct balance of nucleation sites between the nuclear and cytoplasmic faces of the SPB. Together these contributions by CDK inherently link correct SPB morphogenesis, age and fate.

*For correspondence:
slj@stowers.org (SLJ);
ms433@hermes.cam.ac.uk (MS)

†These authors contributed equally to this work

Competing interests: The authors declare that no competing interests exist.

## Introduction

Spindle orientation in self-renewing stem cell divisions exploits structural asymmetries built into the centrosome cycle to create a directional bias that links differential fate with an invariant pattern of age-dependent centrosome inheritance (*Fu et al., 2015*; *Pelletier and Yamashita, 2012*; *Rebollo et al., 2007*; *Venkei and Yamashita, 2018*; *Wang et al., 2009*; *Yamashita et al., 2007*). Perturbation of these mechanistic links impairs self-renewal, prompting an imbalance between stem cell pools and differentiating progeny that disrupts development or causes tumorigenesis (*Gonzalez, 2013*; *Mukherjee and Brat, 2017*; *Vertii et al., 2018*; *Wang et al., 2009*; *Wodarz and Näthke, 2007*).

The premise of an invariant pattern of spindle pole inheritance coupled to spindle orientation in cells dividing asymmetrically first emerged in the budding yeast *Saccharomyces cerevisiae* (*Pereira et al., 2001*), a unicellular organism that divides into a larger mother cell and a smaller daughter cell or bud. In *S. cerevisiae*, all aspects of spindle morphogenesis are controlled by the spindle pole body (SPB), the analog of the animal centrosome (*Byers, 1981*; *Cavanaugh and Jaspersen, 2017*; *Fu et al., 2015*; *Jaspersen and Winey, 2004*; *Winey and Bloom, 2012*). The SPB consists of three major layers — an inner plaque facing the nucleus, a central plaque rooted in the nuclear envelope and an outer plaque facing the cytoplasm (*Figure 1A* and *Figure 1—figure supplement 1*). A specialized extension on one side, the half-bridge, is required for SPB duplication

(*Byers and Goetsch, 1975*). Microtubule (MT) nucleation sites arise by recruitment of the conserved γ-tubulin complex (γTC but also referred to as the γ-tubulin small complex or γ-TuSC) composed of Tub4 (yeast γ-tubulin), Spc98/GCP3 and Spc97/GCP2 (*Farache et al., 2018*; *Geissler et al., 1996*; *Knop, 1997a*; *Kollman et al., 2011*; *Winey and Bloom, 2012*). The inner plaque organizes intranuclear spindle MTs by docking γTCs onto Spc110/pericentrin (*Knop, 1997b*). By contrast, nucleation of cytoplasmic or astral microtubules (aMTs) occurs from two distinct γTC-docking sites set up by the cytoplasmic receptor Spc72/CDK5RAP2, upon binding Nud1/centriolin at the outer plaque or Kar1 at the half-bridge (*Gruneberg, 2000*; *Knop and Schiebel, 1998*; *Lin et al., 2015*; *Pereira et al., 1999*).

Conservative SPB duplication generates a new SPB next to the old SPB inherited from the previous cell cycle, thus laying the foundations for inherent functional asymmetry linked to SPB age. Genetic analyses in combination with electron microscopy (EM) and, more recently, structured illumination microscopy (SIM), have contributed toward a model mapping the ordered addition of individual components according to three landmark events — satellite assembly at the distal end of the bridge, generation of a duplication plaque and a final side-by-side stage with the bridge connecting duplicated SPBs (*Adams and Kilmartin, 1999*; *Burns et al., 2015*; *Cavanaugh and Jaspersen, 2017*; *Jaspersen and Ghosh, 2012*; *Jaspersen and Winey, 2004*; *Rüthnick and Schiebel, 2018*; *Winey and Bloom, 2012*). By onset of SPB separation and spindle assembly, the old SPB has established first contacts with the bud through existing aMTs while the newly assembled SPB delays aMT organization until a ~ 1 µm-long spindle has formed (*Shaw et al., 1997*). This intrinsic functional asymmetry, in interplay with extrinsic cues, primes spindle polarity and orientation along the mother-bud axis in association with the stereotyped inheritance of the old SPB by the bud (*Geymonat and Segal, 2017*). In agreement with live imaging data (*Juanes et al., 2013*; *Segal et al., 2000*; *Shaw et al., 1997*), three-dimensional ultrastructural analyses demonstrate aMT asymmetric organization built into the early stages in the spindle pathway, with aMTs emerging from the old SPB outer plaque and the bridge both at the satellite and side-by-side stages in cells proceeding unperturbed (*Byers and Goetsch, 1975*; *McIntosh and O'Toole, 1999*; *O'Toole et al., 1999*).

Core cell cycle controls linking aMT organization with landmark events along the spindle pathway might involve phosphorylation targets at the SPB. Phosphorylation sites have been identified in many SPB components but their possible significance to intrinsic SPB functional asymmetry remains unknown (*Fong et al., 2018*; *Huisman et al., 2007*; *Keck et al., 2011*; *Lin et al., 2011*; *Lin et al., 2014*; *Rock et al., 2013*). In *S. cerevisiae*, cell cycle progression is controlled by a single cyclin-dependent kinase (CDK), Cdc28/Cdk1, that associates with a series of phase-specific cyclins to control various cell cycle events (*Morgan, 2007*). We have previously implicated the S-phase CDK Clb5-Cdc28 in enforcing aMT asymmetry. Indeed, Clb5 inactivation in the sensitized *cdc28-4* background, *cdc28-4 clb5Δ*, specifically abrogates the delay in aMT organization at the new SPB relative to spindle assembly with concomitant disruption of spindle polarity (*Segal et al., 2000*; *Segal et al., 1998*). More recently we have correlated aMT temporal asymmetry with the acquisition of Spc72 at the new SPB outer plaque during spindle assembly (*Juanes et al., 2013*). However, that study could not determine how Spc72 temporal asymmetry arises or its direct impact on γTC distribution at distinct nucleation sites. Moreover, the idea that intrinsic SPB asymmetry has such structural basis has been brought into question in a recent study, which suggested instead that aMT asymmetry stems solely from extrinsic factors (*Lengefeld et al., 2018*).

Here we undertook detailed analysis of the mode of distribution of Spc72 and components of the γTC throughout the spindle pathway by SIM. Our data show that biased recruitment of Spc72 and the γTC at the old SPB stems from manifest asymmetry throughout $G_1$ and during the side-by-side stage, which is preserved until spindle assembly begins. Our data also show that CDK targets Nud1 in order to enforce Spc72 biased partition during S phase. Moreover, CDK is required for the γTC inner to outer plaque ratio that underlies the normal distribution of nucleation sites at the two faces of the SPB. Finally, CDK also promotes the assembly of the new SPB in the correct order by which γTC addition to the outer plaque always follows the complete assembly of the inner plaque. Taken together, our study reveals key contributions of CDK toward accurate SPB morphogenesis that secure the correct interplay between intrinsic and extrinsic asymmetries that determine spindle polarity and differential fate.

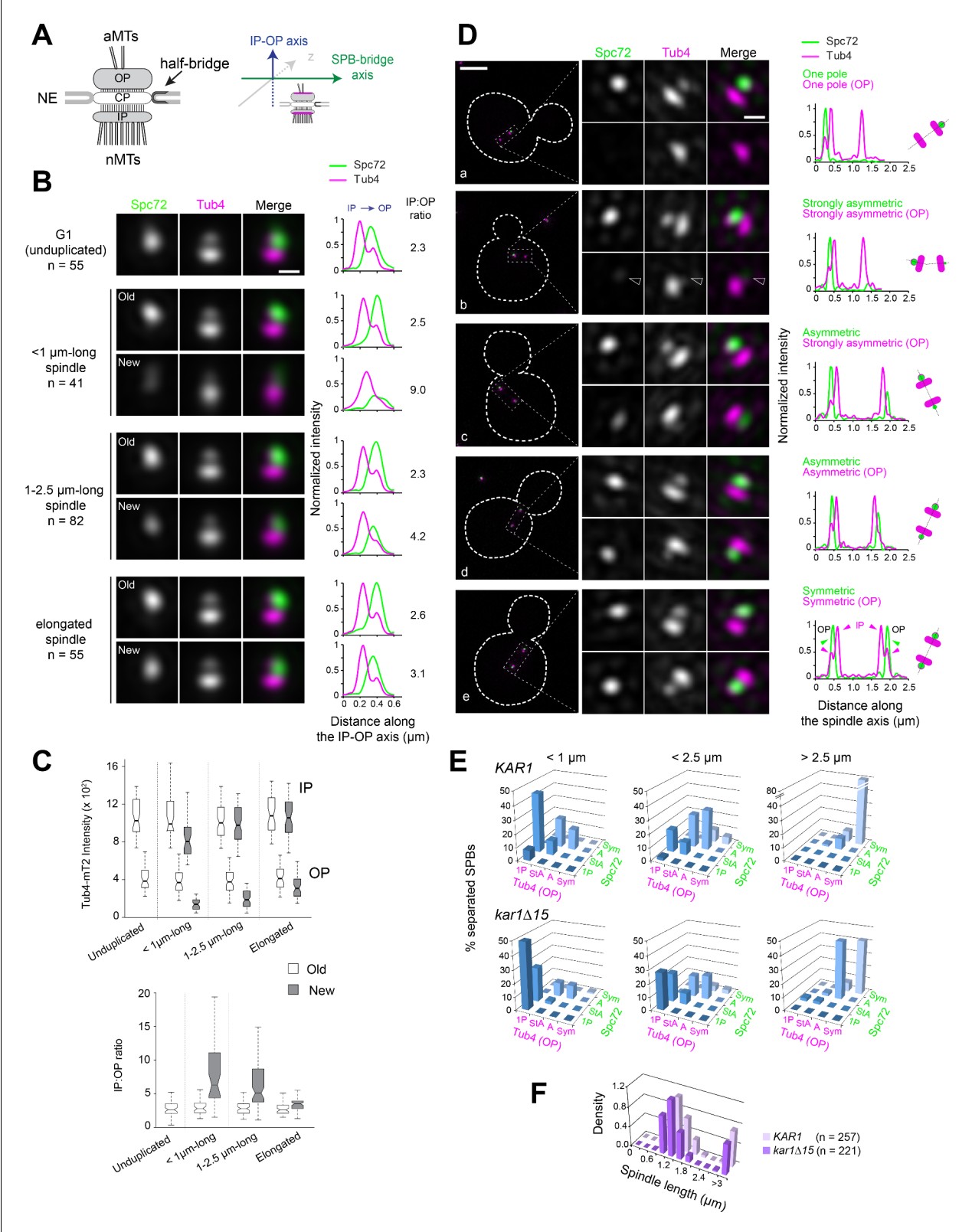

**Figure 1.** Tub4 accumulation at SPB outer plaques follows Spc72 asymmetric recruitment during spindle assembly. (**A**) [Left] Simplified schematic of the SPB showing the inner (IP) and outer (OP) plaques that nucleate nuclear (nMTs) and cytoplasmic or astral microtubules (aMTs), respectively. The central plaque (CP) anchors the SPB in the nuclear envelope (NE). The half-bridge is a modified region of the nuclear envelope involved in SPB duplication; it is also able to nucleate microtubules during $G_1$ phase of the cell cycle (**Byers and Goetsch, 1975**). [Right] To study the distribution of SPB components,

*Figure 1 continued on next page*

Figure 1 continued

SIM images of individual SPBs were aligned using Tub4-mT2 as a reference, as previous work showed that it is present at the inner and outer plaque in different amounts (see *Burns et al., 2015* and *Figure 1—figure supplement 2*). SPBs were assigned to a cell cycle and spindle stage using bud morphology and the distances between SPBs, referred to as SPB inter-distances. (B) Average image from realigned SPBs in each class. Scale bar, 200 nm. The number of SPBs is indicated. Linescan analysis (5-px width) shows the intensity and localization of Spc72-Venus as well as the distribution of Tub4-mT2 at the IP and OP. Linescan intensities were normalized relative to maximal values at the old SPB of elongated spindles. The IP:OP ratio is based on the Tub4-mT2 signal. (C) In addition to averaging, the intensity of Tub4-mT2 at the inner and outer plaques was measured in individual images. Values are plotted by stage and SPB identity (top); IP:OP intensity ratios (bottom) for the same dataset are also shown. Boxplots depict the 5th, 25th, 50th, 75th and 95th centiles. Notches represent 95% CI of the median. (D) Representative SIM images showing the distribution of Spc72-Venus (green) and Tub4-mT2 (magenta) at SPBs of cells with short spindles. Merged images with cell outlines (scale bar, 2 µm) and single channels and merged cropped images for each SPB (scale bar, 200 nm) are shown paired with internally normalized linescan analysis (3-px width) to indicate fluorescence intensity along the spindle axis as represented in the cartoons: (a) Spc72 and Tub4 present at one outer plaque; (b–d) progressive decline of asymmetry for Spc72 followed by Tub4, in (b) a hollow arrowhead points to weak Spc72/Tub4 labels at the new SPB cytoplasmic face; (e) both components symmetric. Modes of label are classified as described in Materials and methods. (E) Quantitation of Spc72-Venus and Tub4-mT2 distribution by spindle stage in *KAR1* and *kar1Δ15* strains. 1P=one pole; StA = strongly asymmetric; A = asymmetric; Sym = symmetric, as described in (D). Note that categories for Tub4 label are based on the relative intensity of the outer plaque foci only. Those correspond to the one (1P) or two (StA, A, Sym) outer peaks of the intensity profile (in magenta) along the spindle axis as indicated by arrowheads for the linescan in (e). By contrast the intensity profile for Spc72 is represented exclusively by one (1P) or two (StA, A, Sym) single peaks (in green). (F) Distribution of spindle lengths in the *KAR1* and *kar1Δ15* asynchronous cell populations analyzed.

The online version of this article includes the following figure supplement(s) for figure 1:

**Figure supplement 1.** Simplified SPB structure.
**Figure supplement 2.** Quantitative detection of γTC by SIM shows SPB size scaling and roles for the outer plaque.
**Figure supplement 3.** Discrimination between SPB side and top view in SIM images.
**Figure supplement 4.** Spc72 and Tub4 asymmetric localization in the W303 yeast background.
**Figure supplement 5.** Spc72 asymmetry in multiple yeast genetic backgrounds.

# Results

## Asymmetric outer plaque structure and γTC partition during spindle assembly revealed by SIM

We have previously proposed that the intrinsic link between SPB age, temporally asymmetric aMT organization and SPB fate is underpinned by Spc72 biased localization to the old SPB during spindle assembly (*Juanes et al., 2013*). Yet, how this asymmetry is built and the associated impact on the localization of γTC at three nucleation sites — inner plaque, outer plaque and the bridge — remain unknown. SIM represents an effective approach to answer these questions as demonstrated by a previous study focused on core molecular events in SPB duplication (*Burns et al., 2015*). Of particular relevance is the ability of SIM to resolve SPB inner and outer plaques shown by that study, using strains expressing endogenous SPB components fused to fluorescent tags. Accordingly, single SPBs of cells expressing Tub4-mTurquoise2 (mT2) showed two distinct foci that could be assigned to inner and outer plaques by co-label with YFP-Spc110 or YFP-Spc72, respectively (*Burns et al., 2015* and *Figure 1—figure supplement 2A*). Two foci were also observed using Spc97-mT2 or Spc98-mT2 (*Burns et al., 2015*). In all cases, the focus co-localized with YFP-Spc110 was more intense than the signal co-localized with YFP-Spc72, consistent with the idea that there is more γTC at the inner plaque of the SPB than at the outer plaque.

To establish the relative ratio of γTC at the inner and outer plaque, we determined the intensity of Tub4-mT2, Spc97-mT2 or Spc98-mT2 foci in individual SIM images (*Figure 1—figure supplement 2B*). Based on the accepted nuclear to cytoplasmic MT ratio of ~5–10 (*Erlemann et al., 2012*), we anticipated a similar ratio for the γTC. Against expectations, quantitative analysis uncovered an overall inner to outer plaque ratio (IP:OP ratio) of ~2–3 for each γTC component. While it is clear from individual measurements there is considerable heterogeneity in γTC that will be further investigated below, this heterogeneity, the lower z-resolution of SIM or other issues with quantitation are unlikely to account for the observed IP:OP ratio. In diploid cells, the number of MTs increases two-fold (*Byers and Goetsch, 1974*). Analysis of Tub4-mT2 distribution in haploids and diploids showed a roughly two-fold increase in γTC at both the inner and outer plaques of diploid SPBs (*Figure 1—*

*figure supplement 2C*). A previous challenge with single color SIM analysis of γTC involved the orientation of the SPB (*Lengefeld et al., 2018*). To overcome this limitation, strains contained second reference label (typically Spc72, Nud1 or Spc110; see Materials and methods) so that images in which inner and outer plaques overlapped vertically (top view) could be excluded. Judging from the apparent overlap between single Tub4 foci and the core outer plaque component Nud1, only ~12% of all SPBs in our SIM images showed a top view (*Figure 1—figure supplement 3*). Collectively, these data point to an unexpected, marked contribution of the outer plaque to bulk γTC content at the SPB and illustrated the utility of SIM for analysis of SPB structure. This finding is consistent with the notion that Tub4 asymmetry observed by wide-field fluorescence microscopy might reflect events at the cytoplasmic face of the SPB (*Juanes et al., 2013*).

The γTC localizes to two cytoplasmic SPB substructures: the outer plaque and the (half-)bridge. To better understand possible γTC heterogeneity and γTC distribution to the outer plaque or half-bridge as contributors to intrinsic SPB asymmetry, we analyzed Tub4-mT2 by SIM in asynchronous cell populations in either *KAR1* or *kar1Δ15* backgrounds. The *kar1Δ15* mutation abrogates Spc72 binding to Kar1 (*Pereira et al., 1999*), thus precluding γTC localization to the bridge. Strains also contained Spc72-Venus (*Figure 1*).

Cell images were grouped by cell cycle/spindle stage (SPB number and spindle length) and SPB age (old versus new) based on measure of inner plaque intensities and/or position within the cell. To compare Tub4 localization among multiple SPBs in a cell cycle/spindle stage, we used computational methods to align dual color SIM images based on Tub4-mT2 fluorescence at the inner and outer plaque of SPBs within that category and used to generate an averaged image (*Figure 1A–B*, see Materials and methods). This type of analysis is advantageous because it shows the likelihood that a protein is present in a given location based on many cells and allows for positional comparison between different proteins (*Burns et al., 2015*). From these averaged images and their quantitation by linescan analysis (*Figure 1B*) or from quantitation of individual SPBs (*Figure 1C*), the structural asymmetry between old versus new SPB was apparent. Unduplicated SPBs from $G_1$-phase cells exhibited a Tub4 IP:OP ratio between 2 and 3 that was also manifest in old SPBs throughout the remaining of the cell cycle. By contrast, the ratio was highest in new SPBs early in spindle assembly but decreased later in the spindle pathway.

Linescan analysis along the spindle axis in maximal SIM image projections of wild type and *kar1Δ15* cells was used to elucidate the contribution of the bridge pool of γTC to its asymmetry (*Figure 1D–F*), validating our data from averaged images while revealing further significant trends. First, Tub4-mT2 was apparent at the inner plaque of the new SPB early in spindle assembly (<1 μm-long spindles), before Spc72-Venus or Tub4-mT2 accumulated at the new SPB outer plaque (*Figure 1D* a versus b-e). Both components were strongly asymmetric at the cytoplasmic face of the SPB in *KAR1* cells or mainly present at the old pole in *kar1Δ15* cells (*Figure 1E*). The *KAR1* and *kar1Δ15* samples analyzed showed similar spindle length profiles (*Figure 1F*) confirming that overall spindle dynamics was not affected by the *kar1Δ15* mutation. The complete assembly of the SPB inner plaque served as a further reference in support of the late acquisition of Tub4 by the new SPB outer plaque. Second, incorporation of Tub4-mT2 onto the new SPB trailed Spc72-Venus, consistent with the idea that Spc72 is needed for γTC recruitment to the outer plaque (*Figure 1E*). Taken together, Tub4 exhibited a marked asymmetry at the cytoplasmic face of the SPB at early stages of the spindle pathway underpinned by Spc72 distribution. Further, the bridge-based localization of γTC plays little role in this asymmetry.

Importantly, Spc72 and Tub4 asymmetry during spindle assembly was also observed in the W303 yeast strain background in both individual images and in averaged images grouped by cell cycle/spindle stage (*Figure 1—figure supplement 4*). Quantitative analysis for relative Tub4 partition between inner and outer plaques by stage and SPB identity confirmed that Tub4 asymmetric recruitment is common to both 15D and W303 strain backgrounds (*Figure 1—figure supplement 4B–C*). Validation of Spc72 asymmetry by wide-field fluorescence microscopy analysis in multiple strains (*Figure 1—figure supplement 5*) indicates that the effects are unlikely related to yeast genetic background and/or SIM. Rather, these results demonstrate that intrinsic SPB structural and functional asymmetry is a general feature of the budding yeast spindle cycle.

## Spc72 and Tub4 asymmetries arise during SPB duplication

Landmark molecular events in SPB duplication have been elucidated by EM studies and, more recently, dissected by SIM (*Adams and Kilmartin, 1999*; *Burns et al., 2015*). In order to understand the source of SPB functional asymmetry at onset of spindle assembly, we focused on characterizing events at the cytoplasmic face of the SPB(s) along the duplication pathway, with emphasis on the distribution and incorporation of Spc72 and γTC, not studied in previous SIM analysis. However, it would be tempting to predict the temporality for assembly of the outer plaque on the basis of prevalent models for SPB duplication (*Figure 2A*; *Winey and Bloom, 2012*).

From asynchronous wild type cells coexpressing mT2-Spc110 (to discriminate inner and outer plaque) and Tub4-Venus, we could identify multiple configurations of γTC in $G_1$ cells. In early $G_1$, unduplicated SPBs contained two Tub4-Venus foci, the strongest corresponding to the inner plaque (*Figure 2A,a*). A subset of unduplicated SPBs exhibited an elongated cytoplasmic signal extending beyond the presumed outer plaque (*Figure 2A,b*); this most likely corresponded to bridge-localized γTC. In later $G_1$, two foci of Spc110 were apparent (*Figure 2A,c–d*) along with a second prominent Tub4 focus at the new inner plaque, consistent with the idea that Spc110 incorporation occurs after SPB insertion into the nuclear envelope and that Spc110 recruits γTC to the SPB inner plaque. Duplicated SPBs remained unseparated in S phase (see *Figure 2—figure supplement 1* for a distribution of distances between unseparated SPB pairs) and had a variable cytoplasmic label of Tub4-mT2 extending from the old SPB to the space between the two inner plaques, as shown in the 1-pixel linescans across inner and outer plaques (*Figure 2A,c–d*). These data suggest that γTC is present on the outer plaque of the old SPB throughout SPB duplication and localizes to varying degrees to the bridge. However, it does not appear to be added to the outer plaque of the new SPB as part of SPB duplication in $G_1$.

To quantify the distribution of γTC at the old SPB, bridge and new SPB, we compared Tub4-Venus to Spc42-mT2 in wild type or *kar1Δ15* cells. Spc42 is the first protein incorporated into the new SPB during its formation in $G_1$ phase, and its levels can be used to evaluate progression through SPB duplication (*Adams and Kilmartin, 1999*; *Burns et al., 2015*; *Rüthnick et al., 2017*). Cells at the satellite or duplication plaque stage were recognized by the presence of duplicated Spc42 signals alongside a single Tub4 inner plaque focus (*Figure 2B,a–b*, hollow arrowhead) or at the side-by-side stage by the presence of duplicated Spc42 signals with two Tub4 inner plaque foci (*Figure 2B, c–e*, hollow arrowheads). During the satellite stage, Tub4 was also observed at the old SPB outer plaque (*Figure 2B,a*, solid arrowhead), or favoring the space between the SPB and satellite with a residual signal often apparent at the old SPB outer plaque (*Figure 2B,b*, big and small solid arrowheads, respectively). After appearance of the new inner plaque (*Figure 2B,c–e*), label continued to distribute between those two cytoplasmic locations (solid arrowheads). Label associated with the new SPB outer plaque was not observed. In *kar1Δ15* cells Tub4 retained localization exclusively at the old SPB outer plaque throughout (*Figure 2B,f–g*, solid arrowhead) demonstrating that localization at the center of the SPB pair in *KAR1* cells represented nucleation sites at the bridge. Thus, Tub4 and by extension γTC, localizes to the old SPB and bridge during SPB duplication. The fact that Tub4 is readily detected at the new SPB inner plaque but not at the outer plaque indicates that its asymmetry between cytoplasmic faces of the old versus new SPB is not due to fluorophore maturation. Further, our data clearly demonstrate that SPB insertion is not the trigger for completion of outer plaque assembly.

As γTC is recruited to the outer plaque and bridge via Spc72, we also examined Spc72 incorporation into the SPB during duplication using Spc42 to mark collectively the satellite stage and duplicated side-by-side SPBs (*Figure 2C*). Like γTC, we observed two localization patterns. Some cells showed single Spc72 label at the old SPB (*Figure 2C,a*, arrowhead) while the majority of cells showed Spc72 label extending from the old SPB to the bridge, with label favoring one or the other location (*Figure 2C,b–d*, big and small arrowheads, respectively) and a small proportion appearing to carry label at the bridge exclusively (*Figure 2C,e*). This dual site distribution was abrogated in *kar1Δ15* cells in which label remained at the old SPB (*Figure 2C,f*). An identical result was obtained when Spc72-Venus localization was determined in reference to Nud1-mT2 label (*Figure 2D*), which also displays two signals from the satellite stage and throughout duplication (*Burns et al., 2015* and *Figure 2—figure supplement 2*). Taken together, these data demonstrated the incomplete

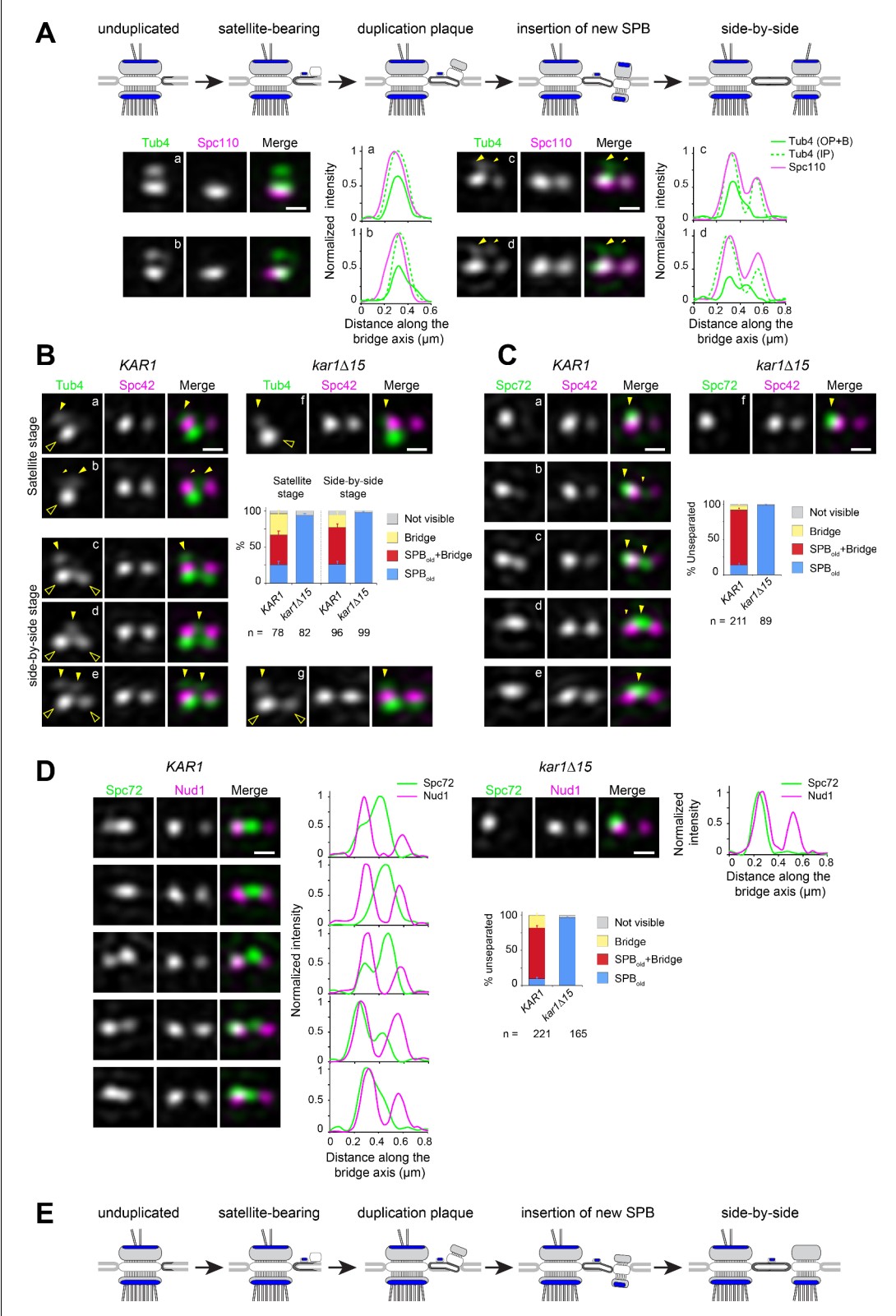

**Figure 2.** Spc72 and γTC are not added to the new SPB during its duplication. (A–D) SIM image projections were obtained after 3D-Gaussian fit and realignment using the indicated reference label. Images were rotated to position the old SPB to the left. Scale bar, 200 nm. (A) (top) Model for landmark events along the SPB duplication pathway deduced by EM and SIM studies (after *Adams and Kilmartin, 1999* and *Burns et al., 2015*). The presumed localization of γTC according to this model is suggested in blue. (bottom) Representative SIM images and corresponding linescan analysis

*Figure 2 continued*

along the inner or outer plaque (1px-width, parallel to the bridge axis, internally normalized) illustrating Tub4-Venus distribution (green) relative to mT2-Spc110 (magenta). Images are ordered along the presumed SPB duplication pathway according to structures seen in early or intermediate steps of SPB duplication leading to the completed side-by-side SPBs: single SPB showing either a single (a) or extended (b) cytoplasmic label; (c–d) side-by-side stage. Big and small yellow arrowheads point to Tub4 at the old SPB outer plaque and bridge, respectively, as described in the text (B–C) Using Spc42-mT2 intensity, SIM images were organized to represent stages of SPB duplication. (B) The distribution of Tub4-Venus (green) localization relative to Spc42-mT2 (magenta) during SPB duplication in *KAR1* (a–e) versus *kar1Δ15* (f–g) cells and plot for distribution of modes of Tub4 localization at the cytoplasmic face of the SPB. Solid yellow arrowheads point to sites on the cytoplasmic face of the SPB hollow arrowheads indicate inner plaque(s) (C) The distribution of Spc72-Venus (green) localization relative to Spc42-mT2 (magenta) during SPB duplication in *KAR1* (a–e) versus *kar1Δ15* (f) cells and plot for modes of Spc72-Venus distribution at unseparated SPBs. Solid arrowheads mark dual localization of Spc72 as described in the text. Error bars, standard error of the proportion. (D) Spc72-Venus (green) and Nud1-mT2 (magenta) were localized in *KAR1* versus *kar1Δ15* cells by SIM. Representative images of cells undergoing SPB duplication, paired to linescan analysis (three px-width along the bridge axis; internally normalized) and plot for distribution of modes of Spc72 localization relative to Nud1. Error bars, standard error of the proportion. (E) New model depicting the temporality for completion of inner and outer plaque assembly relative to other landmark events in the SPB duplication cycle as emerging from the data presented here. In particular, the new SPB outer plaque is incompletely assembled at the side-by-side stage, and thus unable to recruit γTC.

The online version of this article includes the following figure supplement(s) for figure 2:

**Figure supplement 1.** SPB inter-distance as basis for classification of SPBs by cell cycle stage.
**Figure supplement 2.** Modes of Nud1 localization in unseparated SPBs.
**Figure supplement 3.** A subset of cytoplasmic nucleation sites coincides with Kar1 localization in unseparated SPBs.

assembly of the new SPB outer plaque prior to SPB separation as depicted in the new model proposed in *Figure 2E*.

Localization of Spc72 and Tub4 at and away from the bridge during the side-by-side stage was also assessed relative to the bridge component Kar1 (*Figure 2—figure supplement 3*). By contrast to the dual distribution detected by linescan analysis for both Tub4 and Spc72 relative to Spc42 used as reference (*Figure 2—figure supplement 3A–B*), Kar1 was exclusively positioned between unseparated Spc42 signals (*Figure 2—figure supplement 3C* and also see *Burns et al., 2015*; *Seybold et al., 2015*). Colocalization between Kar1 and the central peak of cytoplasmic Tub4 was abrogated in cells expressing instead Kar1Δ15 (*Figure 2—figure supplement 3D*). Three color imaging experiments using Venus-Kar1 in addition to Spc42-mCherry and Tub4-mT2 or Spc72-mT2 confirmed that the central peak of Spc72 or Tub4 co-localized with Kar1 at the bridge in unseparated SPBs (*Figure 2—figure supplement 3E*). Importantly, these experiments clearly showed that a significant fraction of both Spc72 and Tub4 are maintained at the old SPB, even in the presence of bridge-localized pools of each protein.

To confirm that the asymmetric localization of Spc72 and γTC arises during SPB duplication, we examined Spc72-Venus and Spc98-mT2 (another component of the γTC) in *KAR1* and *kar1Δ15* cell populations released from a metaphase block imposed by repression of *CDC20*. This allowed us to follow synchronous populations proceeding unperturbed along the subsequent cell cycle to ensure we captured and staged all modes of localization during early landmarks of the spindle pathway (*Figure 3*). SPB duplication took place in cells with incipient buds (*Figure 3A,a,b,f*), which began SPB separation 15 min later (*Figure 3B-D*). SIM analysis of these synchronized cells confirmed the inherent asymmetry built into the SPB duplication cycle, the overall absence of Spc72 and Spc98 at the new SPB outer plaque, and a modest contribution to Spc72 label at the new SPB attributed to the bridge, as inferred from the nearly absolute asymmetry at onset of SPB separation observed in *kar1Δ15* cells. These findings validated our analysis using asynchronous cell populations (*Figures 1–2*). We additionally quantified Spc72 intensity at the old versus new SPBs upon separation in *kar1Δ15* cells and derived the corresponding asymmetry index (the difference between intensities at old vs. new SPB over the total intensity, see Materials and methods) to determine when Spc72 began to be incorporated to the new SPB outer plaque. As indicated by a significant drop in Spc72 asymmetry index at the transition from the ~1 μm long stage, Spc72 began to accumulate in short spindles prior to anaphase onset (*Figure 3E*).

In conclusion, throughout SPB duplication and the ensuing side-by-side stage, Spc72 and the cytoplasmic pool of γTC are present at the old SPB outer plaque and the bridge but are absent from

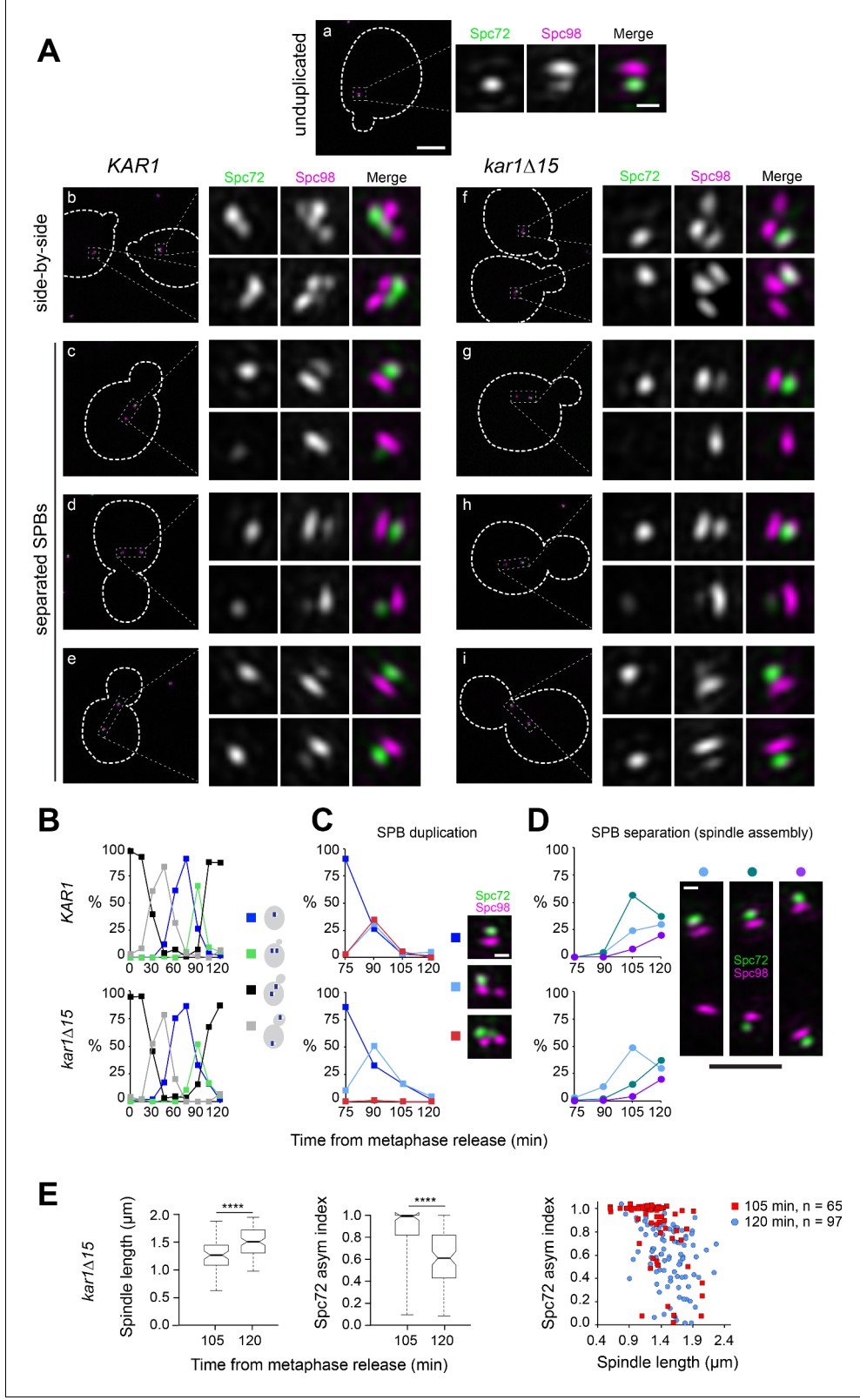

**Figure 3.** Spc72 accumulation at the new SPB outer plaque begins at the ~1 µm-long spindle stage. (**A–E**) *KAR1* or *kar1Δ15* cells co-expressing Spc72-Venus (green) and Spc98-mT2 (magenta) were synchronized in metaphase using $P_{MET3}$-*CDC20*, then released into the cell cycle. (**A**) Representative SIM images showing the distribution of Spc72-Venus (green) and Spc98-mT2 (magenta). Merged images with cell outlines (scale bar, 2 µm) and cropped images

*Figure 3 continued*

of SPBs by channel or merged (scale bar, 200 nm) are shown: (a) unduplicated, (b, f) side-by-side (c–e and g–i), separated SPBs. (B) Cell cycle progression in terms of the spindle pathway. (C–D) The localization of Spc72-Venus and Spc98-mT2 was quantitated at the indicated times in cells with side-by-side SPBs (C; scale bar, 200 nm) or separated SPBs (D; scale bars: white, 200 nm and black, 1 μm) Total number of cells analyzed by time point was 70, 85, 107, 123, 70, 114, 150, 198 and 125 (*KAR1*); 96, 160, 103, 70, 128, 139, 320, 205 and 224 (*kar1Δ15*). (E) Boxplots for distribution of spindle length or Spc72 asymmetry index (left) and scattered plot for Spc72 asymmetry index as a function of spindle length at the indicated time points were generated by 3-D SIM analysis. ****, p<0.0001 according to Mann Whitney two-tailed test.

the new SPB outer plaque. This asymmetry persists at onset of SPB separation with the new SPB acquiring Spc72, and γTC, as cells transit past the ~1 μm-long spindle stage.

## S-phase CDK is required for Spc72 asymmetric recruitment

Our data show that Spc72 (and γTC) bias to the old SPB at onset of spindle assembly stems from inherent asymmetry built into the SPB duplication cycle. The observation that tethering the γTC-binding domain of Spc72 to Cnm67 (a component incorporated to the new SPB from the satellite stage) abrogates the delay in aMT acquisition by the new SPB during spindle assembly (*Juanes et al., 2013*) is consistent with this model. Intrinsic SPB asymmetry is also abolished in *cdc28-4 clb5Δ* mutant cells, with both SPBs exhibiting aMTs at onset of SPB separation (*Segal et al., 2000*). In order to determine whether this was also due to loss of Spc72 asymmetry, asynchronous populations of wild type, *clb5Δ*, *cdc28-4* and *cdc28-4 clb5Δ* mutant cells expressing Spc72-GFP were subject to quantitative imaging analysis. Using Spc42-CFP as a reference, the level of Spc72 at each SPB in cells carrying short spindles was normalized and the Spc72 asymmetry index calculated (see Materials and methods). Short spindles of wild type and *cdc28-4* mutant cells exhibited marked Spc72 asymmetry while a modest decrease was observed in *clb5Δ* cells (*Figure 4A*). By contrast, asymmetry was significantly impaired in *cdc28-4 clb5Δ* cells (*Figure 4A–B*), in agreement with the observation that this mutant nucleates aMTs at both SPBs at onset of spindle assembly (*Segal et al., 2000*).

We further examined unseparated SPBs in these strains by SIM, as Clb5-Cdc28 is active during this stage of the SPB/cell cycle. A strong synergy was observed between the *cdc28-4* and *clb5Δ* mutations — in the double mutant, Spc72 was present at the new SPB already before SPB separation in more than 50% of cells analyzed (*Figure 4C–E*). Additionally, introducing the *kar1Δ15* allele did not affect excess symmetry caused by *cdc28-4 clb5Δ* mutations confirming the involvement of the new SPB outer plaque (*Figure 4F*). The genetic interaction was specific for *clb5Δ* and not shared by a *clb3Δ clb4Δ* mutant even though Clb3 and Clb4 are needed for timely SPB separation (*Juanes et al., 2011*; *Segal et al., 1998* and *Figure 4—figure supplement 1*). The requirement for S-phase CDK in restricting Spc72 to the old SPB during the side-by-side stage was further confirmed by SIM analysis of *KAR1* or *kar1Δ15* cells upon induction of a non-degradable version of the CDK inhibitor Sic1, which prevents entry into S phase and arrests cells prior to SPB separation (*Figure 4G*). Taken together, these data suggest that S-phase CDK imparts correct spindle polarity by enforcing Spc72 asymmetry.

## CDK enforces Spc72 asymmetric recruitment via Nud1

To identify SPB components that might represent direct CDK targets involved in Spc72 asymmetric recruitment, we undertook a candidate-based approach guided by an SPB phospho-proteomic dataset (*Keck et al., 2011*) to generate a series of strains in which single SPB components (Cnm67, Kar1, Spc72 and Nud1) were replaced for mutant versions in which CDK consensus sites (minimal S/T-P or S/T-P-X-K/R) were cancelled by substituting the critical S or T for A (see Materials and methods). The resulting strains were evaluated for Spc72 asymmetry in cells with short spindles by wide-field fluorescence microscopy. From those candidates, only cells expressing Nud1[7A] in which positions $S^{21}$, $S^{294}$, $T^{388}$, $T^{392}$, $S^{469}$, $T^{806}$ and $T^{843}$ were substituted by A to cancel seven CDK consensus sites, prompted a significant loss of Spc72 asymmetry in vivo (*Figure 5A–B*) despite a slight reduction in Nud1 expression level (*Figure 5—figure supplement 1A*). Mutation of these sites abolished phosphorylation by purified Clb5-Cdc28-as in vitro (*Figure 5—figure supplement 1B*). Kinase specificity

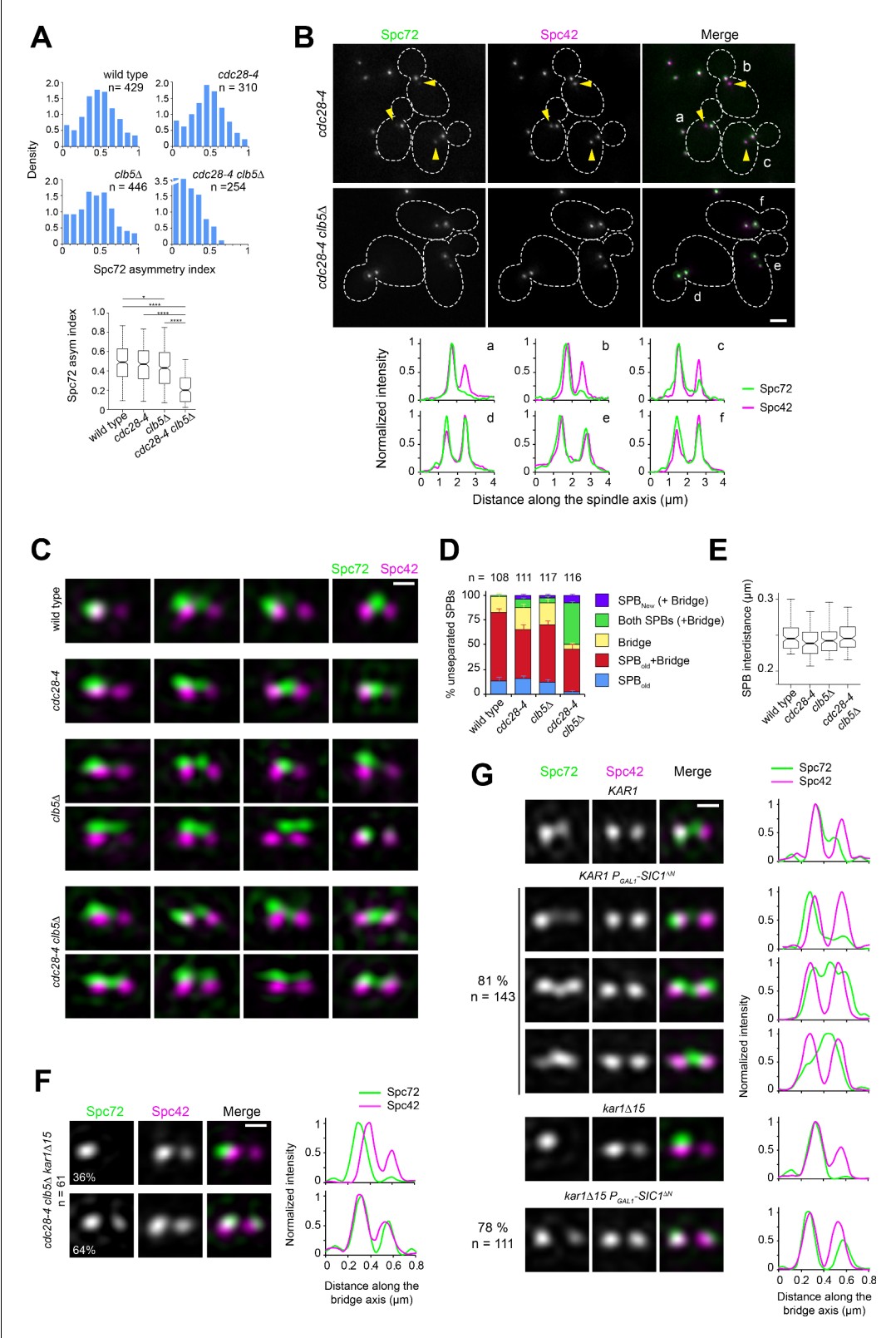

**Figure 4.** Spc72 asymmetric recruitment requires S-phase CDK. (**A–B**) The distribution of Spc72-GFP (green) and Spc42-CFP (magenta) was compared in wild-type, *clb5Δ*, *cdc28-4* and *cdc28-4 clb5Δ* cells grown at 23°C. (**A**) Histograms show the distribution of Spc72 asymmetry index in cells with short spindles (<2.5 μm length) for each sample. Boxplots below depict the 5th, 25th, 50th, 75th and 95th centiles. Notches represent 95% CI of the median. *p=0.0189 and ****p<0.0001, according to Kruskal-Wallis and Dunn's multiple comparison tests. (**B**) Representative wide-field fluorescence images and
*Figure 4 continued on next page*

Figure 4 continued

corresponding linescan analysis along the spindle axis showing Spc72 asymmetry in *cdc28-4*, but not *cdc28-4 clb5Δ* cells. Yellow arrowheads point to new SPBs weakly labelled by Spc72-GFP, underscoring asymmetry. Scale bar, 2 μm. (C–D) SIM analysis of unseparated SPBs of the indicated strains. (C) Representative 3D-realigned images of unseparated SPBs labeled by Spc72-Venus (green) and Spc42-mT2 (magenta). Loss of asymmetry occurred in *cdc28-4 clb5Δ* cells in which Spc72 recruitment at the new SPB outer plaque began prior to SPB separation. Scale bar, 200 nm. (D) Quantitation of modes of Spc72 localization in unseparated SPBs of the indicated strains. Error bars, standard error of the proportion. (E) Boxplot showing the distance between Spc42 foci in all cells tallied. 5th, 25th, 50th, 75th and 95th centiles are shown. Notches represent 95% CI of the median. (F) Representative images for the two modes of Spc72 localization observed in unseparated SPBs of *cdc28-4 clb5Δ kar1Δ15* asynchronous cells. Images were subject to realignment based on the Spc42-mT2 label. The corresponding linescans for fluorescence intensity along the bridge axis (3-px width; internally normalized) are also shown. (G) *KAR1* or *kar1Δ15 P_{GAL1}-SIC1^{ΔN}* cells (or control cells carrying a vector instead of pLD1) containing Spc72-Venus (green) and Spc42-mT2 were grown in synthetic-raffinose medium to early log phase at 25°C and induced at the same temperature by addition of 3% galactose. After 1.5 hr induction, SIM images were acquired and the SPBs in small-budded cells analyzed. Uniform cell cycle arrest was observed after 3.5 hr. Representative SIM images of Spc72-Venus (green) and Spc42-mT2 (magenta) are shown with corresponding linescans (3-px width; internally normalized). Scale bar, 200 nm.

The online version of this article includes the following figure supplement(s) for figure 4:

**Figure supplement 1.** S phase CDK inactivation anticipates Spc72 recruitment at the new outer plaque in unseparated SPBs.

was confirmed by the use of the purified analog-sensitive CDK (*Bishop et al., 2000*), with five of the seven sites also confirmed in subsequent global phosphoproteome studies of the SPB (*Fong et al., 2018*; *Rock et al., 2013*). Accordingly, Nud1^{7A} exhibited reduced bulk phosphorylation relative to wild type Nud1 in vivo, based on mobility shift assessed by western blot analysis of protein extracts from unperturbed cells or cells depleted of Cdc5 (*Figure 5—figure supplement 1A and C*, see Materials and methods).

If CDK-dependent phosphorylation acts through Nud1 to promote SPB functional asymmetry, then we would predict that *nud1^{7A}* cells would impair SPB inheritance, and display Spc72 symmetry at early stages of the spindle pathway, along the lines observed for a *cdc28-4 clb5Δ* mutant. Consistent with this idea, expression of Nud1^{7A} increased the percentage of cells in which the new SPB entered the bud (*Figure 5C*). To determine an underlying link to Spc72, *nud1Δ* strains complemented by *NUD1* or *nud1^{7A}* and coexpressing Spc72-Venus and Spc98-mT2 were analyzed by SIM. *nud1^{7A}* cells already exhibited a marked reduction in Spc72 and Spc98 asymmetry at onset of SPB separation (*Figure 5D–F*). Furthermore, we observed Spc72 recruitment to the new outer plaque in side-by-side SPBs of *nud1^{7A}* cells (*Figure 5G*). While reminiscent of that observed in *cdc28-4 clb5Δ* cells, the phenotype appeared less penetrant in *nud1^{7A}* cells (*Figure 5G–H*), suggesting that while Nud1 may be a key target, other CDK substrates are probably involved.

To investigate the basis for CDK control of Spc72 asymmetry via Nud1, the ability of Spc72 to bind purified Nud1 or Nud1^{7A} after phosphorylation by yeast CDK was determined in vitro. CDK treatment resulted in a nearly 10-fold increase in pull-down efficiency for wild type Nud1 but exerted no effect on the extent of Spc72 binding to Nud1^{7A} (*Figure 5I*). The interaction and its enhancement by CDK phosphorylation were retained by a Nud1 domain extending between amino acid positions 250–600 (*Figure 5—figure supplement 1D–E*). The binding domain defined here overlapped but was distinct from that identified in a previous study by yeast two-hybrid analysis using the C-termini of Spc72 and Nud1 (*Gruneberg, 2000*).

We hypothesized that a phospho-dependent change in affinity might promote structural asymmetry if CDK activity inherently favored the old SPB outer plaque. To explore a mechanistic basis for such possibility, the interaction between G$_1$/S cyclins and Spc72 was tested. Both Cln2 and Clb5 as part of CDK complexes were efficiently pulled down by purified Spc72 in vitro (*Figure 5J*). It follows that changes in affinity induced by CDK phosphorylation of Nud1 may enforce Spc72 retention at the old SPB within a temporal window of opportunity initiated by Spc72 itself, thus reinforcing its bias to the old SPB.

## Orderly SPB assembly and correct γTC inner to outer plaque ratio require CDK

Spc72 was already present at the new SPB outer plaque in greater than 50% of unseparated SPBs of asynchronous *cdc28-4 clb5Δ* cells, a pronounced advancement compared to the recruitment of Spc72 to the new SPB outer plaque only part way along spindle assembly observed in wild type

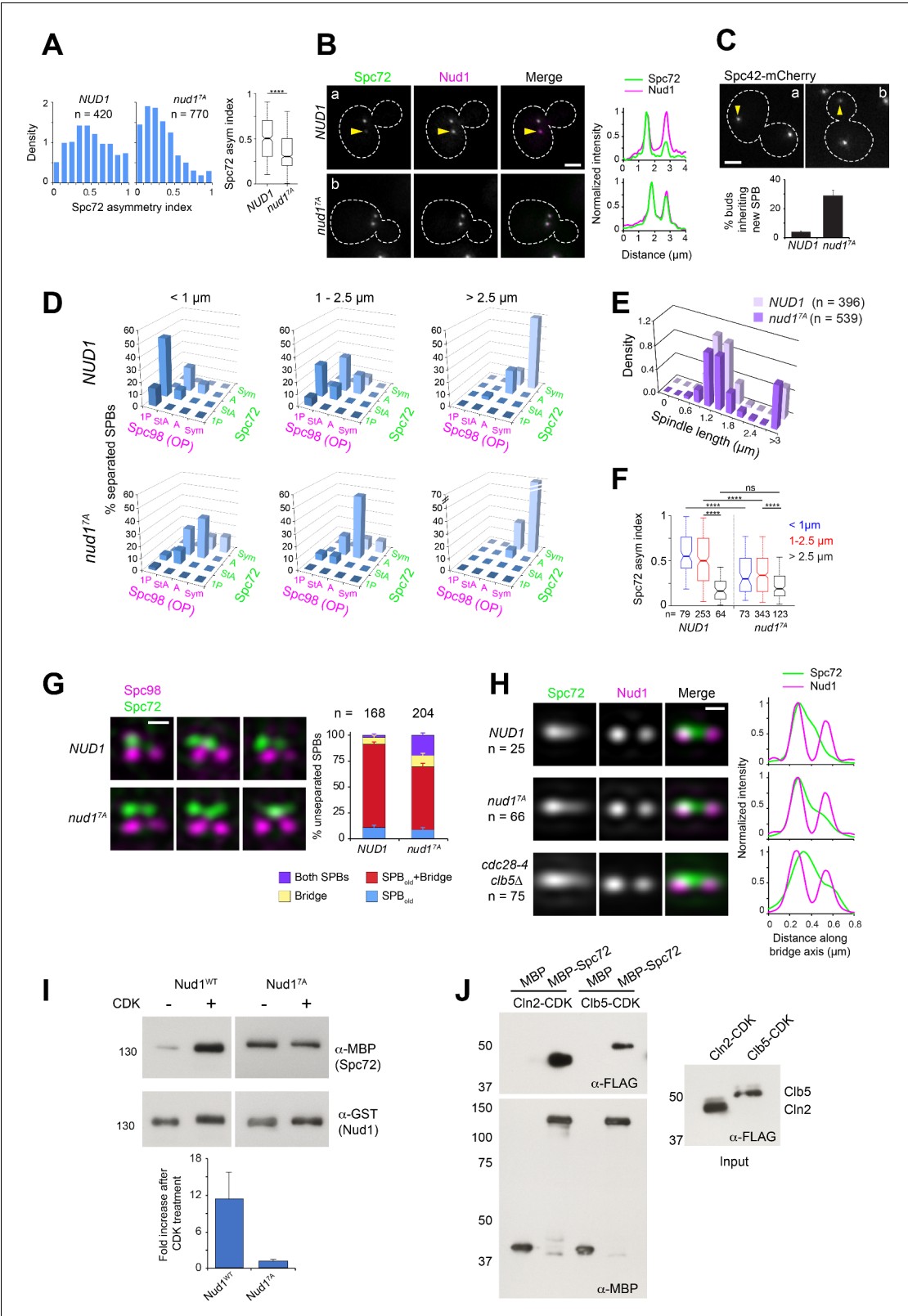

**Figure 5.** Substitutions cancelling CDK phosphorylation in Nud1 result in loss of Spc72 asymmetry. (A–B) Cells expressing wild type Nud1-CFP or Nud1[7A]-CFP along with Spc72-GFP were analyzed by wide-field fluorescence microscopy. (A) Histogram showing the distribution of Spc72 asymmetry index in each sample. ****p<0.0001 according to the Mann Whitney two-tailed test. (B) Representative fluorescence images and corresponding linescan analysis along the spindle axis. Yellow arrowheads highlight asymmetric localization pointing to weaker label at the new SPB. Scale bar, 2 µm. (C)

*Figure 5 continued on next page*

Figure 5 continued

Proportion of *NUD1* versus *nud1^7A* cells in which the new SPB was inherited by the bud. Mean of three counts of 200 cells is shown, error bars SD. (D) Distribution of Spc98-mT2 and Spc72-Venus by spindle stage was quantitated as described in *Figure 1*. (E) The spindle length distribution within the cell populations is shown. (F) Boxplot analysis showing Spc72 asymmetry index by spindle stage. 5th, 25th, 50th, 75th and 95th centiles are shown. Notches represent 95% CI of the median. ****, p<0.0001 according to Kruskal-Wallis and Dunn's multiple comparison tests. (G) (Left) Representative SIM images from cells with side-by-side SPBs to compare the distribution of Spc72 in *NUD1* versus *nud1^7A* cells. In *nud1^7A* cells, Spc72-Venus (green) was incorporated into the SPB outer plaque in a fraction of cells with side-by-side SPBs. Bar, 200 nm. (Right) Quantification of modes of Spc72 recruitment in unseparated SPBs of *NUD1* versus *nud1^7A* cells. Error bars, standard error of the proportion. (H) Comparison of the extent of disruption of Spc72 asymmetry in *nud1^7A* versus *cdc28-4 clb5Δ* cells assessed in averaged images of unseparated SPBs from asynchronous cell populations, compiled after 3D-realignment in reference to Nud1 label. The corresponding linescans for fluorescence intensity along the bridge axis (3-px width; internally normalized) are shown. (I) Effect of CDK phosphorylation of Nud1 vs Nud1^7A on Spc72 binding in vitro. (Top) A mixture of Cln2 and Clb5-CDK was used to phosphorylate GST-Nud1 or GST-Nud1^7A bound to beads. After extensive washing to terminate the kinase reaction, beads were incubated with identical amounts of purified MBP-Spc72 for 2 hr followed by repeated washes. Eluted fractions were resolved in SDS-PAGE and the presence of Nud1 and Spc72 determined by western blot analysis. (Bottom) Plot representing average fold increase in Nud1-Spc72 binding upon CDK treatment from three independent experiments. Error bar, SD. (J) Ability of MBP-Spc72 to pull down Cln2 or Clb5 as part of CDK complexes in vitro. An equivalent amount of input (right, 5 s exposure) was analyzed under identical conditions to the pull-down (left, 20 s exposure).

The online version of this article includes the following figure supplement(s) for figure 5:

**Figure supplement 1.** Nud1 and Nud1^7A characterization in vivo and in vitro.

cells. The *nud1^7A* mutation also advanced Spc72 recruitment in unseparated SPBs, yet to a lesser extent (*Figure 5*), suggesting that additional CDK targets might be implicated. Yet, in view of the marked penetrance in *cdc28-4 clb5Δ* cells, it was of great interest to quantify the impact of CDK inactivation on the establishment of nucleation sites by analyzing Tub4-Venus localization in reference to Spc42-mT2 in *cdc28-4 clb5Δ* cells as done before for wild type cells (*Figure 2*). Surprisingly, this analysis pointed to comparable intensities of Tub4-Venus at inner versus outer plaque in this mutant (*Figure 6—figure supplement 1A–B*). We therefore proceeded to study Tub4-Venus localization relative to Nud1-mT2 instead, to assign inner and outer plaques at all stages without relying solely on the premise that the inner plaque corresponded to the more intense of the two Tub4 foci in a given SPB.

To determine the earliest point of γTC recruitment at the new SPB outer plaque in *cdc28-4 clb5Δ* cells, the localization of Tub4-Venus was assessed in 3D-realigned SIM images of unseparated SPBs according to the Nud1-mT2 foci. Prior to the side-by-side stage, (paired Nud1-mT2 foci with a single Tub4 inner plaque focus), analysis of individual images indicated that Tub4-Venus at the cytoplasmic face distributed between the old SPB outer plaque (*Figure 6A,a–b*; yellow arrowheads) and the bridge in wild type cells while present exclusively at the old SPB outer plaque in *kar1Δ15* cells (*Figure 6A,c–d*, yellow arrowheads), in agreement with data presented in *Figure 2*. By contrast, 5 out of 12 *cdc28-4 clb5Δ* cells displayed a new Tub4 signal that colocalized with Nud1 at the presumptive new outer plaque before Tub4 presence at the new inner plaque (*Figure 6A,e* versus *Figure 6A, f-gA*; hollow versus solid white arrowhead). A similar proportion of *cdc28-4 clb5Δ kar1Δ15* cells showed Tub4 at the new outer plaque in the absence of an inner plaque label (*Figure 6A,h* versus *Figure 6A,i-j*). These data suggested that *cdc28-4 clb5Δ* cells might complete the assembly of the new SPB outer plaque before the new inner plaque is fully established, a reversal of the wild type sequence of events. Realigned images of cells after reaching the side-by-side stage (Tub4 at both inner plaques) were averaged for quantitation (*Figure 6B*). Wild type *KAR1* or *kar1Δ15* cells subject to this protocol confirmed the intrinsically asymmetric side-by-side structure as documented in *Figure 2* and *Figure 2—figure supplements 2* and *3*. Yet, *cdc28-4 clb5Δ* (*KAR1* or *kar1Δ15*) mutant cells exhibited both significant accumulation of Tub4 at the new SPB outer plaque and an overall reduction in Tub4 IP:OP ratio at both SPBs. By analyzing individual images used for these averages, the advanced Tub4 incorporation at the outer plaque relative to the inner plaque at the side-by-side stage was further apparent (*Figure 6—figure supplement 1C*). This advanced Tub4 addition at the outer plaque before inner plaque complete assembly followed Spc72 recruitment (*Figure 6—figure supplement 1D*).

Contrary to what was observed in *cdc28-4 clb5Δ* cells, Tub4-Venus inner to outer plaque ratio in *nud1^7A* cells was essentially unaffected (*Figure 6—figure supplement 2A*). Although the *nud1^7A* mutation perturbed Tub4 asymmetry at the side-by-side stage in line with the advanced recruitment

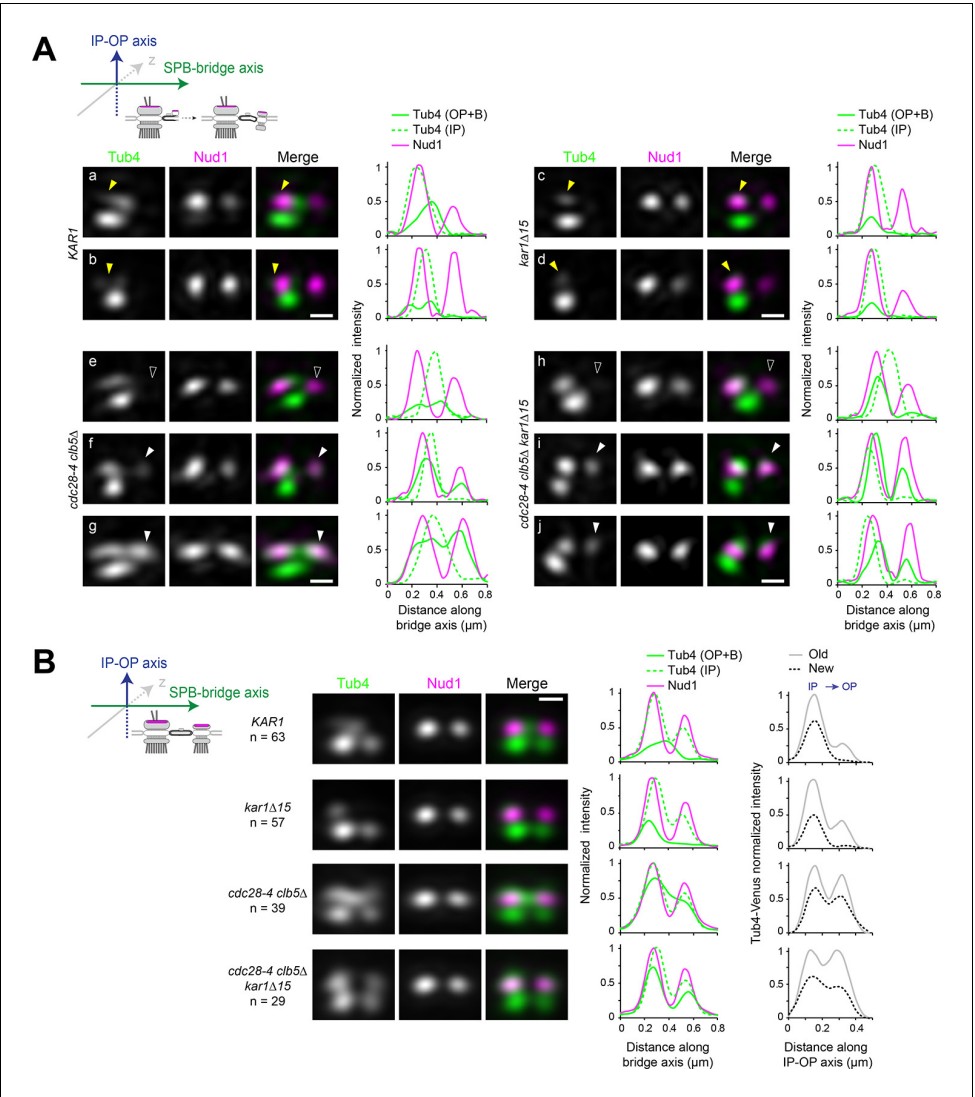

**Figure 6.** Tub4 IP:OP ratio and the sequence for new SPB assembly are perturbed in *cdc28-4 clb5Δ* cells. (**A–B**) SIM analysis of the indicated strains expressing Tub4-Venus and Nud1-mT2 as a reference for the outer plaque. Cartoons indicate the stage in each case and label used for 3D-realignment. Scale bar, 200 nm. (**A**) Individual SIM images and linescan analysis (1-px width; internally normalized) for the indicated strains corresponding to unseparated SPBs prior to nucleation establishment at the new SPB inner plaque, realigned with respect to Nud1-mT2. Tub4 colocalized partly with Nud1 at the old SPB in wild type cells (**a-b**; yellow arrowhead) or exclusively in *kar1Δ15* cells (**c-d**; yellow arrowhead). By contrast, in *cdc28-4 clb5Δ* and *cdc28-4 clb5Δ kar1Δ15* cells at this stage, Tub4 additionally colocalized with the new Nud1 focus (compare (**e**) with (**f-g**) or (**h**) with (**i-j**); hollow versus solid white arrowheads). (**B**) Averaged images compiled after 3D-realignment with respect to Nud1-mT2 of side-by-side SPBs. Linescans for fluorescence intensity along the bridge and IP-OP axis are also shown (1-px width; internally normalized).

The online version of this article includes the following figure supplement(s) for figure 6:

**Figure supplement 1.** Tub4 IP:OP ratio and sequence of Tub4 recruitment during SPB duplication in *cdc28-4 clb5Δ* cells.

**Figure supplement 2.** Nud1[7A] anticipates Spc72 recruitment without otherwise affecting the Tub4 IP:OP ratio.

of Spc72 at the new SPB outer plaque (*Figure 5*), it was insufficient to reverse the normal sequence for addition of Tub4 recruitment during assembly of the new SPB (*Figure 6—figure supplement 2B*).

Taken together, inactivation of S-phase CDK in *cdc28-4 clb5Δ* cells not only abrogated asymmetry otherwise arising from the SPB duplication cycle but also reversed the orderly sequence of events required to build the new SPB, with concomitant impact on the balanced recruitment of the γTC to each SPB face. These data demonstrated two separable contributions of CDK to the control of new SPB assembly impacting intrinsic asymmetry and correct SPB morphogenesis. First, CDK helps retain Spc72 at the old SPB outer plaque thus enforcing asymmetry in unseparated SPBs, at least in part, via phosphorylation of Nud1. Second, CDK further imposes the sequential addition and the ensuing distribution of γTC at the inner and outer plaques, most likely, by phosphorylation of other SPB components (see Discussion). These findings point to critical links between CDK temporal control, correct SPB morphogenesis and functional asymmetry.

## Discussion

### SIM reveals that maturation and acquisition of nucleation capability are compartmentalized and proceed separately at inner and outer plaques of the new SPB

Historically, SPB assembly is thought to begin with the formation of the satellite, followed by addition of outer plaque components, including Nud1 and, perhaps by extrapolation, Spc72 and the γTC (*Adams and Kilmartin, 1999*; *Rüthnick and Schiebel, 2016*; *Winey and Bloom, 2012*). Because mutants that disrupt SPB insertion into the nuclear envelope (i.e., *mps2-1*, *ndc1-1*) form a complete outer plaque that is competent for aMT nucleation, SPB insertion is not required for outer plaque assembly (*Winey et al., 1991*; *Winey et al., 1993*). Yet, our current work challenges the idea that completion of the new outer plaque, specifically addition of Spc72 and γTC, temporally precedes SPB insertion and even separation in wild type cells. Rather, these distal components are added later as the new SPB matures part-way along spindle assembly.

Previous SIM analysis focused on short spindles was unable to estimate the overall proportion of SPBs with γTC at the outer plaque, in part, because a fraction of those SPBs were presumably aligned perpendicularly to the imaging plane (*Lengefeld et al., 2018*). By contrast, our use of a second label and 3-D realignment allowed us to analyze virtually all SPBs in every cell cycle stage to uncover outer plaque asymmetry. Furthermore, unlike direct stochastic optical resolution microscopy (dSTORM) in *Lengefeld et al., 2018*, which presumed a model for MT formation in vivo based on in vitro work, our analysis made no assumptions and only took into account fluorescence intensity — a reliable readout of γTC abundance given the increase observed as a function of cell ploidy or during the cell cycle. In agreement with data presented here, studies in *A. nidulans* and *S. pombe* revealed heterogeneity in γTC levels at SPBs throughout interphase compared to more uniform levels in mitosis (*Bestul et al., 2017*; *Gao et al., 2019*). Differences in the composition of γTC and its interactor, Mozart, as well as phosphoregulation of Mto2 (an activating subunit of the Spc72 ortholog, Mto1), are thought to underlie SPB asymmetry in these cases (*Borek et al., 2015*; *Gao et al., 2019*). However, all of these factors are absent in budding yeast.

EM and fluorescence microscopy studies show that the inner plaque nucleates 5 to 10-fold more MTs than the outer plaque in *S. cerevisiae* (*Erlemann et al., 2012*). Therefore, we anticipated a correlative 5–10 IP:OP ratio in γTC levels rather than the ~2 fold difference measured. This apparent discrepancy between γTC levels and MT nucleation suggests that the availability of γTC is not the sole regulator of MT formation in the cell. Further work will help elucidate the roles that the outer and inner plaque receptors, Spc72 and Spc110 respectively, the MT polymerase Stu2 (the budding yeast Dis1/TOG family member) and various post-translational modifications play in differential MT nucleation on each face of the SPB (*Fong et al., 2018*; *Gunzelmann et al., 2018*; *Huisman et al., 2007*; *Keck et al., 2011*; *Lin et al., 2011*; *Lin et al., 2014*). In conclusion, our work illustrates an additional mode of regulation for aMT organization involving the addition of Spc72 and γTC to the new SPB with a temporality emerging, at least in part, from CDK-dependent phosphorylation of Nud1.

### CDK contributions to the SPB cycle and intrinsic asymmetry

CDKs promote multiple events along the spindle pathway (*Avena et al., 2014*; *Byers and Goetsch, 1974*; *Chee and Haase, 2010*; *Elserafy et al., 2014*; *Fitch et al., 1992*; *Haase et al., 2001*;

*Huisman et al., 2007*; *Jaspersen et al., 2004*; *Jones et al., 2018*; *Juanes et al., 2011*; *Liang et al., 2013*; *Lin et al., 2014*; *Rüthnick and Schiebel, 2016*; *Segal et al., 2000*; *Winey and Byers, 1993*) beginning with the requirement for $G_1$ cyclin (Cln)-Cdc28 in SPB duplication. Most *cdc28ts* alleles block cells at the satellite-bearing stage at the restrictive temperature. In turn, *cdc4ts* alleles (that prevent onset of Clb-dependent CDK activation) arrest cells after SPB duplication at the side-by-side stage (*Byers and Goetsch, 1974*). A number of SPB targets have been implicated but the precise molecular details are unknown (*Jaspersen et al., 2004*; *Jones et al., 2018*; *Keck et al., 2011*). CDK activity also ensures that SPB duplication occurs once per cell cycle via Sfi1 — one of the best understood CDK molecular paradigms centered on an SPB component (*Avena et al., 2014*; *Elserafy et al., 2014*). Otherwise, most CDK phosphorylation sites in SPB components studied so far have proven dispensable for viability (e.g. *Huisman et al., 2007*; *Jaspersen et al., 2004*; *Lin et al., 2014*, this study), with synergism observed when CDK site mutants for multiple SPB components are combined (*Jones et al., 2018*). Now, phospho-proteomic datasets convey an integral view of CDK phosphorylation at the SPB (*Fong et al., 2018*; *Keck et al., 2011*; *Lin et al., 2011*; *Rock et al., 2013*) against a backdrop for potential interplay with several cell cycle, and possibly other, protein kinases. However, it remains an ongoing challenge to link this complex regulatory landscape to function or any spatial restrictions tied into SPB age.

We previously implicated Clb5-Cdc28 in enforcing intrinsic SPB asymmetry based on the phenotype of *cdc28-4 clb5Δ* mutant cells, a genetic setup designed to circumvent cyclin functional redundancy (*Segal et al., 2000*; *Segal et al., 1998*). Here, our data indicate that Clb5-Cdc28 is required for timing Spc72 addition to the new SPB outer plaque. Indeed, *cdc28-4 clb5Δ* mutants exhibited a marked advancement of Spc72 recruitment in unseparated SPBs, a defect also apparent in cells overexpressing non-degradable Sic1. Thus, despite any functional overlap between Clns and Clb5, Cln-Cdc28 might be insufficient to sustain SPB intrinsic asymmetry. However, we cannot exclude that both CDK complexes might contribute in wild type context, given that loss of Spc72 asymmetry arises from the strong genetic interaction between *cdc28-4* (a $G_1$ hypomorph *CDK* allele) and *clb5Δ*.

Substitutions impeding CDK phosphorylation of Nud1 also advanced Spc72 recruitment to the new SPB outer plaque leading to substantial symmetry in short spindles. To understand the molecular link between Nud1 phosphorylation by CDK and Spc72 bias, the impact of CDK on the interaction between Nud1 and Spc72 was investigated. Spc72 pull-down by phosphorylated wild type Nud1 as bait was enhanced ~10 fold over an unphosphorylated control. By contrast, CDK could not stimulate binding when Nud1[7A] was used as bait. This change in affinity might translate into structural asymmetry if CDK activity were to inherently favor the old SPB outer plaque. Supporting this notion, both Cln2 and Clb5 as part of CDK complexes interacted with purified Spc72 in vitro. These data suggest that CDK might preferentially associate with the old SPB, which contains Spc72 in $G_1$, to dictate intrinsic asymmetry (*Figure 7*). CDK phosphorylation of Nud1 once Clb5-Cdc28 becomes active, retains Spc72 at the old SPB, while Spc72 acquisition at the new SPB is inherently delayed due to reduced affinity toward unphosphorylated Nud1. Then, delayed acquisition of Spc72 by the new SPB sets in motion a positive feedback loop centered on Spc72 and CDK phosphorylation of Nud1, accelerating outer plaque assembly as the spindle forms. Retention of Spc72 by the old SPB has been demonstrated by *Lengefeld et al., 2018* although the mechanistic basis of how this might be achieved was not discussed. Such a mechanism might effectively establish SPB identity on the basis of Spc72 inherited by the old SPB from the preceding cell cycle. In the absence of CDK phosphorylation of Nud1, however, Spc72 retention by the old SPB would be compromised during duplication and the side-by-side stage when it is partly redeployed to the bridge, and thus become readily shared by both SPBs. We therefore propose a built-in interplay between Spc72-driven recruitment of CDK at the old SPB and the ensuing enhancement of Spc72 binding affinity through Nud1 phosphorylation by CDK as a landmark event in the SPB duplication cycle linking SPB identity and age (*Figure 7*).

Nud1 plays dual roles as an SPB outer plaque component. In addition to docking Spc72, Nud1 is a scaffold protein in the Mitotic Exit Network (MEN), a signaling pathway required for progression across the M/$G_1$ phase boundary (*Gruneberg, 2000*; *Rock et al., 2013*). Previous studies suggested Nud1 as a putative CDK and Cdc14 target, the latter a phosphatase typically reversing CDK phosphorylation (*Bloom et al., 2011*; *Park et al., 2008*). Nud1 is also phosphorylated by Mps1, Cdc5 and Cdc15 (*Keck et al., 2011*; *Maekawa et al., 2007*; *Park et al., 2008*; *Rock et al., 2013*). Nud1 MEN function has been implicated in correct SPB inheritance by linking the activation of the

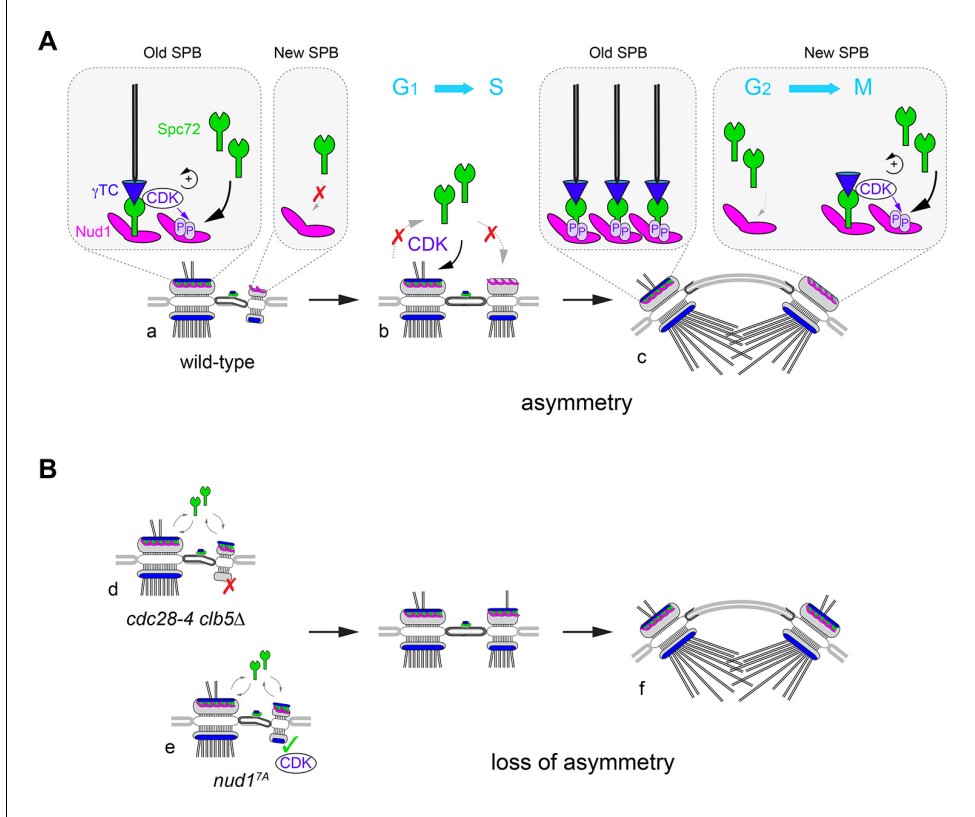

**Figure 7.** A model for CDK control of intrinsic SPB asymmetry. Our data suggest that CDK phosphosites in Nud1 increase affinity toward Spc72. (**A**) Because the old SPB inherits Spc72 and Nud1 from the preceding cell cycle (**a**), CDK recruitment to the old SPB via Spc72 promotes phosphorylation of Nud1 that may reinforce Spc72 retention in wild type cells. De novo Nud1 added to the new SPB (from the satellite state) is presumably not phosphorylated and would have low affinity for Spc72. As a result (**b**), S-phase CDK enforces structural asymmetry during the side-by-side stage. By onset of spindle assembly (**c**), low-affinity binding of Spc72 at the new SPB outer plaque may redirect CDK to phosphorylate Nud1 and trigger the phospho-dependent increase in affinity toward Spc72 that accelerates completion of the new SPB outer plaque. Thus, CDK recruitment via Spc72 translates phospho-dependent affinity between Spc72 and Nud1 into intrinsic structural asymmetry. This built-in delay in new outer plaque maturation manifests in temporal asymmetry in γTC recruitment and aMT organization. (**B**) In the absence of Nud1 phospho-dependent control, Spc72 is no longer restricted to the old SPB during the side-by-side stage with concomitant disruption of intrinsic asymmetry. The more severe phenotype observed upon loss of S-phase CDK implicates further substrates (see text). In *cdc28-4 clb5Δ* cells (**d**), Spc72 symmetry is compounded by mis-regulation of the inner plaque allowing for the incorporation of γTC to the new SPB in the reverse order - first at the outer plaque and then at of the inner plaque. By contrast (**e**), loss of Nud1 phosphosites in *nud1^7A* cells, decreases Spc72 asymmetry without otherwise perturbing inner plaque function. Order of γTC addition is retained but acquisition by the outer plaque is advanced as it precedes SPB separation. In both mutants, short spindles display significant loss of outer plaque structural asymmetry (**f**).

downstream MEN kinase Dbf2 at metaphase with polarized marking of the old SPB by Kar9 (*Hotz et al., 2012*). Yet, later *Campbell et al., 2019* demonstrated that Dbf2 activation is confined past anaphase onset and plays no role in Kar9 control, raising the possibility that the defective inheritance reported in a *nud1* mutant related instead to the role of Nud1 demonstrated here. Supporting this idea, *Gordon et al., 2006* also implicated both Nud1 and Spc72 in control of SPB inheritance during meiosis.

Relative to a *cdc28-4 clb5Δ* mutant, *nud1^7A* exhibited a less penetrant disruption of intrinsic SPB asymmetry. Thus, CDK may act through further substrates at the SPB, and more broadly via other protein kinases such as Mps1 and Cdc5, both dependent on CDK for their activity (*Jaspersen et al., 2004*; *Mortensen et al., 2005*; *Rodriguez-Rodriguez et al., 2016*). CDK also negatively regulates Cdc15 (*Jaspersen and Morgan, 2000*; *König et al., 2010*). Finally, Clb5-Cdc28 might antagonize

Cdc14 by direct phosphorylation (*Li et al., 2014*), while the interplay between CDK thresholds and a constellation of phosphatases acting at the $G_1$/S boundary is beginning to emerge (*Ariño et al., 2019*; *Martín et al., 2020*).

Surprisingly, inactivation of S-phase CDK not only disrupted Spc72 asymmetry but also decreased the γTC IP:OP ratio at SPBs. Moreover, recruitment of γTC during SPB duplication was markedly misregulated — Tub4 often appeared at the new outer plaque before its incorporation to the new inner plaque, a reversal of the normal sequence of events. By contrast, nud1[7A] cells displayed excess symmetry without otherwise altering the order of γTC recruitment at the two faces of the new SPB or the Tub4 IP:OP ratio (*Figure 7*). These observations point to CDK acting via additional targets such that acquisition of γTC normally starts at the new SPB inner plaque. The γTC assembles in the cytoplasm and nuclear import is mediated by a nuclear localization sequence in Spc98 (*Pereira et al., 1998*). Clb5-Cdc28 phosphorylation of Spc110 at the inner plaque is critical for efficient γTC recruitment in early S phase (*Huisman et al., 2007*; *Lin et al., 2014*). Once γTC is docked at the inner plaque (but not at the outer plaque), Spc98 undergoes Mps1 and CDK-dependent phosphorylation (*Pereira et al., 1998*), which is not essential for γTC recruitment per se but might still promote retention and nucleation activity. Thus, CDK phosphorylation prompts maturation of the new inner plaque ahead of the outer plaque. Tub4 also undergoes CDK and Cdc5-dependent phosphorylation that might be important for microtubule organization (*Lin et al., 2011*). In cdc28-4 clb5Δ cells, the combined effect of advanced Spc72 association with the new outer plaque and impaired retention of γTC by the inner plaque through hypophosphorylation of Spc110, Spc98 and Tub4 might cause roughly concurrent binding of γTC to both faces of the SPB with concomitant decrease in the IP:OP ratio. By contrast, the nud1[7A] mutation decreases Spc72 asymmetry without otherwise affecting inner plaque function and IP:OP ratio. In conclusion, CDK-enforced orderly SPB assembly ensures both correct SPB morphogenesis and age-dependent structural asymmetry.

## Intrinsic versus extrinsic control of astral microtubule asymmetry

Temporally asymmetric aMT organization is a hallmark of spindle polarity in yeast —with the pole initially carrying aMTs, the old SPB, becoming the bud-ward pole (*Pereira et al., 2001*; *Segal et al., 2000*; *Shaw et al., 1997*). By contrast to our proposal that age-dependent aMT asymmetry is intrinsically built into the SPB duplication cycle, *Lengefeld et al., 2018* proposed that aMT asymmetry *is exclusively driven* by extrinsic cues, in particular, Kar9.

Extrinsic asymmetries function in two key processes promoting spindle polarity: aMT capture by the polarity determinant Bud6 at the bud cell cortex and Kar9-dependent delivery of aMT plus ends to the bud along actin cables (*Geymonat and Segal, 2017*). However, both extrinsic determinants are necessary but insufficient to link SPB age, identity and fate without intrinsic SPB asymmetry (*Cepeda-García et al., 2010*; *Huisman et al., 2004*; *Juanes et al., 2013*). Conversely, intrinsic SPB asymmetry cannot translate into asymmetric fate unless aMTs from the old SPB are targeted to the bud cell cortex through extrinsic cues. Indeed, in bud6Δ or kar9Δ mutants, aMT temporal asymmetry during spindle assembly *is still manifest* (and therefore intrinsic) but no longer determines the pole ending in the bud (*Yeh et al., 2000*). Furthermore, aMTs are differentially stabilized by cell compartment through those same cortical cues: interaction between Bim1 (yeast EB1) at aMT plus ends and cortical Bud6 stabilizes cortical contacts (*Ten Hoopen et al., 2012*) and aMTs are stabilized during Kar9-mediated transport toward the bud (*Huisman et al., 2007*). Together these precedents explain the correlation between aMT asymmetric dynamics and subcellular localization attributed to cortical cues by *Lengefeld et al., 2018* without otherwise disproving that aMT asymmetry arises from the intrinsic structural asymmetry demonstrated here in multiple strain backgrounds.

The interplay between spatial control of distribution of the nucleation machinery and cell cycle-dependent phosphorylation may turn out a recurrent theme in centrosome control and beyond given the conservation of these nucleation systems and the protein kinase regulatory networks promoting maturation (*Fu et al., 2015*; *Tovey and Conduit, 2018*). Whether these mechanisms prove instrumental for cell cycle control of centrosome asymmetries associated with differential cell fate remains an exciting prospect.

# Materials and methods

## Key resources table

| Reagent type (species) or resource | Designation | Source or reference | Identifiers | Additional information |
|---|---|---|---|---|
| Gene (*Saccharomyces cerevisiae*) | *NUD1* | Saccharomyces Genome Database | SGD:S000005900 | |
| Gene (*Saccharomyces cerevisiae*) | *SPC72* | Saccharomyces Genome Database | SGD:S000000045 | |
| Strain, strain background (*S. cerevisiae*) | 15DaubA | *Richardson et al., 1989*. doi:0092-8674(89)90768-X [pii] | | |
| Strain, strain background (*S. cerevisiae*) | W303 | Saccharomyces Genome Database | | |
| Strain, strain background (*S. cerevisiae*) | BY4741 | Saccharomyces Genome Database | | |
| Strain, strain background (*S. cerevisiae*) | YEF473 | Bi and Pringle, 1996. doi:10.1128/mcb.16.10.5264 | | |
| Strain, strain background (*E. coli*) | Rosetta 2 cells | Novagen (Merck) | # 71400 | |
| Strain, strain background (*S. cerevisiae*) | MGY70 | *Geymonat et al., 2009*. doi: 10.1007/978-1-60327-993-2_4. | | |
| Strain, strain background (*S. cerevisiae*) | MGY139 | *Geymonat et al., 2009*. doi: 10.1007/978-1-60327-993-2_4. | | |
| Antibody | anti-HA 12CA5 (Mouse monoclonal) | Roche | #11583816001 RRID:AB_514505 | WB (1:1000) |
| Antibody | anti-α tubulin B-5-1-2 (Mouse monoclonal) | Sigma (Merck) | T5168 RRID:AB_477579 | WB (1:1000) |
| Antibody | anti-GST 3G10/1B3 (Mouse monoclonal) | Abcam | ab92 RRID:AB_307067 | WB (1:1000) |
| Antibody | anti-MBP (Mouse monoclonal) | New England Biolabs | E8032 RRID:AB_1559730 | WB (1:1000) |
| Chemical compound, drug | paraformaldehyde EM Grade | Ted Pella | # 18501 | |
| Software, algorithm | FIJI | https://fiji.sc/ | RRID:SCR_002285 | |
| Software, algorithm | Plugins for FIJI | http://research.stowers.org/imagejplugins | | |
| Other | Dako mounting media | Agilent Technologies | # S3023 | |

## Yeast strains and genetic procedures

Yeast strains used in this study (Source Data 1) were derived from 15DaubA (*Cepeda-García et al., 2010*; *Juanes et al., 2011*; *Juanes et al., 2013*; *Richardson et al., 1989*) or W303 (*Burns et al., 2015*), except for additional strain backgrounds analyzed in *Figure 1—figure supplement 5*. Standard yeast genetic procedures and media were used throughout (*Guthrie and Fink, 1991*). Synthetic medium with supplements, or rich medium (YEP) contained 2% w/v dextrose, 3% w/v raffinose or galactose when indicated.

Strains carrying the *cdc28-4* allele in combination with deletions in cyclin genes have been previously described (*Segal et al., 1998*). Replacement of endogenous *KAR1* by the *kar1Δ15* allele (*Pereira et al., 1999*) was carried out by one-step disruption using pKS-kar1Δ15-URA3 or pKS-

kar1Δ15-HIS2 (*Juanes et al., 2013*). Deletion of *NUD1* was carried out by one-step disruption using a targeted *KANMX* cassette produced by polymerase chain reaction (*Juanes et al., 2013*). Wild type *NUD1* gene or a mutated version in which seven CDK consensus sites were cancelled using nested PCRs (introducing S/T to A substitutions at positions 21, 294, 388, 392, 469, 806 and 843; referred to in the text as *nud1^7A*) fused to CFP or HA$_3$ were inserted as EcoRI-HindIII fragments into YIplac211 or YIp204, respectively. YIp211-NUD1-CFP or YIp211-NUD1$^{7A}$-CFP were used for transformation of heterozygous *NUD1/nud1::KANMX* diploid cells. Linearization using StuI targeted integration at *URA3*. Following sporulation and tetrad dissection, haploid cells in which the *NUD1* deletion was rescued by the integrative constructs were selected. Haploid cells carrying YIp204-NUD1-HA$_3$ or YIp204-NUD1$^{7A}$-HA$_3$ integrated at *TRP1* in the background of the *NUD1* deletion were similarly constructed. A similar strategy was implemented to construct strains expressing wild type or phospho-site mutant versions of Spc72-GFP, HA$_3$-Kar1 or Cnm67-HA$_3$ in the respective deletion backgrounds. Mutations encoded S/T to A substitutions at positions 232, 243, 552 and 566 of Spc72, positions 149, 216, 222 and 294 of Kar1 and positions 17, 72, 89, 103, 120, 121, 146 and 147 of Cnm67.

Endogenous tagging to produce C-terminal fusions to GFP, YFP, CFP, mCherry, or mTurquoise2 of Tub4, Spc72, Nud1, Spc42, Cnm67 and Spc98 in 15DaubA was carried out using tagging vectors previously described (*Juanes et al., 2011*; *Juanes et al., 2013*; *Ten Hoopen et al., 2012*). N-terminal tags fused to Kar1 and Spc110 were created by single-step replacement with recombinant cassettes targeted to the endogenous loci. W303-derived strains expressing fusions to fluorescent tags have been described in *Burns et al., 2015*.

Strains transformed with pP$_{MET3}$-CDC20-LEU2 (a gift from Ethel Queralt) were used for synchronization by metaphase arrest and release as follows. Early log cell cultures grown in synthetic medium lacking methionine at 23°C were arrested by addition of 2.5 mM methionine in order to repress *CDC20* and incubation was continued for 3 hr at 23°C. Uniform arrest was verified by microscopy. Cells were then harvested by centrifugation at 2000 rpm for 5 min at room temperature, rinsed twice and released by resuspending into synthetic medium lacking methionine at 23°C. Aliquots were collected every 15 min and processed for SIM. For inducible expression of non-degradable Sic1Δ$^{aa2-50}$ under the *GAL1* promoter, cells were transformed with pLD1 (*Noton and Diffley, 2000*) linearized with ApaI. Mid-log cells grown in raffinose-containing synthetic medium at 23°C were induced by addition of 3% galactose and aliquots removed and processed every half hour. Uniform arrest (until cells exhibited single elongated buds) was achieved after 3.5 hr. P$_{MET3}$-CDC5 cells expressing Nud1-HA$_3$ or Nud1$^{7A}$-HA$_3$ at endogenous levels were grown to early log in synthetic medium without methionine at 30°C. Cultures were divided and their volume doubled with identical fresh medium without or supplemented with 2 mM methionine final concentration, respectively. After 3.5 hr at 30°C, uniform arrest due to depletion of Cdc5 was confirmed by microscopy analysis in the cultures supplemented with methionine. Cells were harvested, rinsed with ice-cold water and processed for protein extraction and western blot analysis. Depletion of Cdc5 using this construct causes a uniform arrest with a MEN phenotype (large budded cells with fully elongated spindles).

## Wide-field fluorescence microscopy imaging and analysis

Images of cells coexpressing Spc72-GFP and Spc42-CFP (as reference) were acquired with a Nikon Eclipse E800 with a CFI Plan Apochromat 100x, NA 1.4 objective, a Chroma Technology CFP/YFP filter set and a Coolsnap-HQ CCD camera (Roper Scientific) (*Guo and Segal, 2017*) as five-plane Z-stacks of fluorescence images at a distance of 0.8 μm between planes with 2 × 2 binning, paired to a DIC image at the middle focal plane (*Juanes et al., 2013*). Stacks were processed into 2-color overlays of maximal intensity 2-D projections and analyzed with MetaMorph software (Molecular Devices). Linescans for fluorescence intensity along the spindle axis were generated with the line tool set to 3-pixel width and normalized intensity plots produced in Microsoft Excel. Integrated intensities were determined in a 7 × 7 pixel region and cell background subtracted for each channel. An asymmetry index was calculated as the absolute difference between the relative values at each SPB (i.e. Spc72/Spc42 intensity ratio) divided by the sum of the same values. The index ranged from 0 (absolute symmetry) to 1 (absolute asymmetry). Statistical analysis was carried out using GraphPad Prism and the open source package R. Boxplots depict 5th, 25th, 50th, 75th and 95th centiles with notches indicating the 95% confidence interval of the median.

## SIM imaging and analysis

Preparation of cell slides for SIM and image acquisition were according to *Burns et al., 2015*. Cells from early-log cultures grown at 22°C in synthetic complete medium supplemented with 0.1 g/L adenine were fixed in 4% paraformaldehyde (Ted Pella) in 100 mM sucrose for 15 min at room temperature and washed twice in phosphate-buffer saline, pH 7.4. An aliquot of cells was resuspended in Dako mounting media (Agilent Technologies, #S3023), placed on a cleaned number 1.5 coverslip, covered with a cleaned glass slide then allowed to cure overnight at room temperature.

SIM images were acquired with an Applied Precision OMX BLAZE (GE Healthcare) equipped with an Olympus 60 × 1.42 NA Plan Apo oil objective. Images were collected in sequential mode with two or three PCO Edge sCMOS cameras (Kelheim, Germany) for each acquisition channel. Color alignment from different cameras in the radial plane was performed using the color alignment slide from GE Healthcare. In the axial direction, color alignment was performed using 100 nm TetraSpeck beads (ThermoFisher, F8803). Reconstruction was accomplished with the softWoRx v6.52 software (GE Healthcare) according to manufacturer's recommendations with a Wiener filter of 0.001. In most cases images are YFP/mT2 with 514 nm excitation for YFP and then 445 nm excitation for mT2. In some cases, we modified the protocol for mT2/YFP/mCherry acquisition with the mCherry acquired first and excited with the 568 nm laser. The dichroic in every case was 445/514/561 with emission filters at 460–485 nm, 530–552 nm and 590–628 nm for mT2, YFP and mCherry, respectively.

Three-dimensional analysis of SIM images was carried out using custom macros and plugins for the open source program FIJI (*Schindelin et al., 2012*). Plugins and source code are available for download at http://research.stowers.org/imagejplugins. First, spots corresponding to unseparated or separated SPB pairs were manually identified based on a reference label (typically Spc42 or Nud1) and sorted for analysis of the query label according to the experiment. If the maximum SPB intensity was in the first or last slice for either query or reference label, the SPB was excluded from further analysis. Next, for unseparated SPB pairs, the two spots in the reference channel representing the SPB were fitted to two 3D-Gaussian functions and realigned along the axis between these functions using [jay_sim_fitting_macro_multicolor_profile.ijm]. The higher intensity spot was assigned as the old SPB (*Burns et al., 2015*; *Unruh et al., 2018*). This realignment allowed the distribution of signal in the query channel to be analyzed against a single point of reference. For analysis of SPBs containing components of the γTC as the reference channel, the inner and outer plaque signal were selected, with the stronger signal assigned to the inner plaque (see Figure S1). These images were fitted to two 3D Gaussian functions and subject to realignment. In some cases, after realignment, images were averaged and scaled as described previously (*Burns et al., 2015*), using [merge_all_-stacks_jru_v1.ijm] then [stack_statistics_jru_v2.ijm]. Reconstructed images were scaled 4 × 4 with bilinear interpolation and presented as max projections in z over the relevant slices. After adjusting consistently brightness and contrast, and scaling to 8-bit image depth, cell images were incorporated into final figures using Adobe Illustrator.

SPB-SPB distances (referred to as SPB inter-distances) within cells were calculated after Gaussian fitting of the reference signal. Arbitrary boundaries along the spindle pathway were set according to the distribution of SPB inter-distances along the cell cycle measured in reference asynchronous cell populations labelled with Spc42-mTurquoise2 (*Figure 2—figure supplement 1*) as follows:<0.35 µm, unseparated SPBs;<1 µm-long spindles; 1–2.5 µm-long spindles;>2.5 µm elongated spindles. Additionally, distance distributions were assessed for conformity in each experiment. In the special case of analyses of γTC components with respect to the outer plaque component Spc72, SPBs were visually inspected and those in which inner and outer plaques overlapped vertically (as indicated by overlapping Spc72 and the sole γTC signals) were excluded from analysis, due to limited resolution in the z axis to assess inner and outer plaque signals. Similar considerations were applied when Tub4 label was quantified instead in reference to the core outer plaque component Nud1 in order to exclude SPBs presented in top view (*Figure 6—figure supplement 1C* and *Figure 6—figure supplement 2*). As shown in *Figure 1—figure supplement 3* (using Nud1 as reference to include both old and new SPBs in this analysis), only ~12% of SPBs fell into this category. When indicated, an asymmetry index was derived from SIM data as the absolute difference between the intensity values of label at each SPB divided by the sum of the same values.

Linescan analysis along the IP-OP axis or the SPB-bridge axis was performed at 1–3- or 5-px width dependent on the experiment. When stated, normalization was carried out internally between

maximal and minimal intensities in individual images or with respect to the indicated reference image in a set. Spindles were arbitrarily classified (*Figure 1D–E* and *Figure 5C*) according to the difference in label intensity between the SPBs as follows: 'one pole', label visible at one outer plaque only; 'strongly asymmetric', at least an 8-fold difference in label intensity between SPBs; 'asymmetric', between 8-fold and 1.3-fold difference; 'symmetric', less than 1.3-fold difference.

To directly compare Tub4 intensity in haploid and diploid cells, a haploid Tub4-mT2 Spc110-YFP strain was mixed with a Tub4-mT2/Tub4-mT2 diploid prior to imaging. After spot-fitting and realignment, images were assigned to the haploid or diploid group based on Spc110-YFP signal. Data from individual SPBs and averaged SPBs is shown.

## Protein production and purification

Recombinant MBP fusions to Nud1 or Nud1[7A] were expressed from pMAL-c4X (New England Biolabs). Induction in Rosetta 2 cells (Novagen) was triggered with 0.5 mM IPTG followed by incubation at 16°C overnight. Cells were then harvested by centrifugation 10 min at 5000 rpm at 4°C and washed with ice-cold $H_2O$. Frozen cell pellets were stored at −80°C. Bacterial cells were lysed by sonication and MBP fusions were purified onto amylose resin (New England Biolabs) as previously described (*Ten Hoopen et al., 2012*). To elute MBP-fusion proteins, the resin-bound fraction was resuspended in 500 µl of elution buffer (20 mM TRIS pH 7.5, 200 mM NaCl, 1 mM EDTA, 10 mM mercaptoethanol + 30 mM Maltose) and incubated on a roller for 5–15 min at 4°C. This was repeated and successive eluates tested for protein presence. Fractions were pooled together and subject to dialysis in HKEG buffer (20 mM HEPES, 50 mM KCl, 1 mM EDTA, 5% Glycerol) using Maxi GeBaFlex-tube (Generon) at 4°C, concentrated by evaporation at 4°C and protein concentration evaluated by SDS-PAGE analysis against a BSA standard curve. Protein aliquots were stored frozen at −80°C.

Protein expression and purification from yeast was based on an auto-selection expression system previously developed (*Geymonat et al., 2009*). Cln2[ΔNt] and Clb5[Δdb], both stabilized versions of the cyclins Cln2 and Clb5, respectively (*Hadwiger et al., 1989*; *Jacobson et al., 2000*), were produced as GST or Twin-Strep-tag fusion proteins in the context of overexpressed Cdc28/Cks1. Briefly, PCR cassettes encoding the mutant cyclins along with gapped pMG1 or pMG6 expression vectors were used to transform the yeast expression host MGY70 (*MAT a ura3-1 trp1-28 leu2Δ0 lys2 his7 mob1::kanR pep4::LEU2 [URA3-MOB1]*) or its derivative MGY139 (*MATα ura3 trp1 his3 leu2 lys2Δ0 pep4::LYS2 mob1::kanR cdc28::LEU2 [URA3-MOB1-CDC28]*) by gap-repair. Auto-selection for the expression construct is achieved by passage of the resulting transformants onto FOA medium to select for loss of the resident *MOB1/CDC28*-carrying plasmid. Proteins were induced in YEP-galactose (1% final) for 8 hr at 30°C. Cells were harvested and proteins purified on glutathione beads according to *Geymonat et al., 2009* or on Strep-Tactin XT resin (IBA) following manufacturer's instructions. CDK protein complexes were eluted with 20 mM reduced glutathione or 50 mM biotin, respectively and dialyzed overnight at 4°C in buffer A (Tris pH 7.5 20 mM, NaCl 150 mM, DTT 0.5 mM, glycerol 10%). Co-purification of Cdc28 and Cks1 was confirmed by SDS-PAGE and cyclins quantified by Coomassie blue staining against a BSA standard curve. Constructs encoding GST-Nud1 or GST-Nud1[7A] were also created by gap-repair of the pMG1 backbone in the yeast host MGY70. The resin-bound fraction was stored in 50% v/v glycerol at −20°C. MBP-Spc72 was expressed from the vector pMG3 in strain MGY853. MBP-Spc72 was eluted with 20 mM maltose and dialyzed o/n at 4°C in buffer A. Eluted proteins were quantified and stored at – 80°C.

## Western blot analysis, kinase and binding assays

Expression level of Nud1-HA$_3$ and Nud1[7A]-HA$_3$ was determined by western blot analysis of whole cell extracts as previously described (*Ten Hoopen et al., 2012*) using monoclonal antibody 12CA5 (Roche) at 1:1000 dilution and monoclonal antibody B-5-1-2 (Sigma) at 1:1000 dilution to detect α-tubulin as loading control.

4 µg of MBP-Nud1 or MBP-Nud[7A] were incubated 30 min at 30°C in the presence of 40 µM ATP, 2 µCi [γ-$^{32}$P] ATP (3000 mCi/mmol; 10mCi/ml, Perkin-Elmer) and 19 ng of purified Strep-tag-Clb5[Δdb]/Cdc28-as in kinase buffer (50 mM Tris pH 7.5, 10 mM MgCl$_2$ and 1 mM DTT) in a final volume of 20 µl. When indicated, Cdc28-as was inhibited by addition of 5 µM 1NM-PP1. Reactions were stopped by addition of 5 µl of 5x Laemmli buffer followed by incubation at 95°C for 3 min. 10

µl- aliquots of phosphorylation reactions were analyzed on a 4–12% gradient SDS-PAGE gel and blotted to a PVDF membrane followed by autoradiography.

Equal amounts (~3–5 µg) of GST-Nud1 and GST-Nud1$^{7A}$ bound to beads (or Nud1$^{(250-600)}$ and Nud1$^{4A(250-600)}$) were incubated for 1 hr at 37°C in a 200 µl reaction containing Strep-tag Clb5$^{\Delta db}$ and/or Cln2$^{\Delta Nt}$/Cdc28/Cks1 (0.5–1 µg) and 1 mM ATP in kinase buffer (50 mM Tris pH 7.5, 10 mM MgCl$_2$ and 1 mM DTT). Mock reactions were carried out in a 200 µl reaction containing kinase buffer only. Reactions were stopped by 3 washes of the beads with ice-cold washing buffer (50 mM Tris pH 7.5, 250 mM NaCl, 0.2% NP-40, 1 mM DTT). Beads were then resuspended in 300 µl of binding buffer (50 mM Tris pH7.5, 250 mM NaCl, 0.2% NP-40, 1 mM DTT, 1% BSA) containing 15 µg of purified MBP-Spc72 and incubated at 4°C for 2 hr on a roller. Beads were washed five times with washing buffer and bound proteins were eluted with 20 mM reduced glutathione. Eluted proteins were analyzed on a 6% SDS-PAGE gel followed by a western blot using monoclonal anti-GST (Abcam) or anti-MBP (New England Biolabs) at 1:1000 dilution to detect GST-Nud1 and MBP-Spc72, respectively. Since MBP-Spc72 and GST-Nud1 have approximate molecular mass of 130 kDa, eluted material was loaded in duplicate gels for western blot analysis and probed with α-MBP or α-GST antibodies, respectively.

## Acknowledgements

We are grateful to Mark Winey for sharing data prior to publication and to Mark Longtine, John Kilmartin and Ethel Queralt for their gift of strains and constructs. We thank Jennifer Gardner, Shannon Burns, Luisa Capalbo, Alexandra Van Hall-Beauvais and Miriam Scarpa for contributing data to this project and Nikola Dzhindzhev for fruitful discussions. Research reported in this publication was supported, in part, by the Stowers Institute for Medical Research and the NIH-NIGMS under award number R01GM121443 (to SLJ), the Department of Genetics, University of Cambridge (to MS) and by CSC Cambridge International Scholarships (to QP and ZG). Original data underlying this manuscript can be downloaded from the Stowers Original Data Repository at http://www.stowers.org/research/publications/LIBPB-1526. The authors declare no competing financial interests.

## Additional information

### Funding

| Funder | Grant reference number | Author |
| --- | --- | --- |
| National Institute of General Medical Sciences | RO1GM121443 | Sue L Jaspersen |
| CSC Cambridge International Scholarship | | Qiuran Peng Zhiang Guo Marisa Segal |

The funders had no role in study design, data collection and interpretation, or the decision to submit the work for publication.

### Author contributions

Marco Geymonat, Conceptualization, Formal analysis, Supervision, Validation, Investigation, Visualization, Methodology, Writing - original draft, Writing - review and editing; Qiuran Peng, Conceptualization, Resources, Formal analysis, Funding acquisition, Validation, Investigation, Visualization, Methodology, Writing - original draft; Zhiang Guo, Conceptualization, Formal analysis, Funding acquisition, Validation, Investigation, Visualization, Methodology; Zulin Yu, Conceptualization, Resources, Data curation, Software, Validation, Methodology, Writing - original draft; Jay R Unruh, Conceptualization, Software, Formal analysis, Supervision, Validation, Investigation, Methodology, Writing - original draft; Sue L Jaspersen, Marisa Segal, Conceptualization, Data curation, Formal analysis, Supervision, Funding acquisition, Validation, Investigation, Visualization, Methodology, Writing - original draft, Project administration, Writing - review and editing

## Author ORCIDs

Marco Geymonat (iD) https://orcid.org/0000-0002-8792-0517
Qiuran Peng (iD) http://orcid.org/0000-0002-4761-0944
Jay R Unruh (iD) https://orcid.org/0000-0003-3077-4990
Sue L Jaspersen (iD) https://orcid.org/0000-0001-8312-7063
Marisa Segal (iD) https://orcid.org/0000-0003-1848-9388

## Decision letter and Author response

Decision letter https://doi.org/10.7554/eLife.59222.sa1
Author response https://doi.org/10.7554/eLife.59222.sa2

# Additional files

## Supplementary files

- Supplementary file 1. Yeast strains.
- Transparent reporting form

## Data availability

Original data will be available upon publication at http://www.stowers.org/research/publications/LIBPB-1526 For phosphoproteomic datasets cited in this manuscript, complete details are included in the citation and reference list. The datasets are accessible as supplemental material at the journal sites.

The following dataset was generated:

| Author(s) | Year | Dataset title | Dataset URL | Database and Identifier |
|---|---|---|---|---|
| Geymonat M, Peng Q, Guo Z, Yu Z, Unruh JRJ, Jaspersen SL, Segal M | 2020 | Orderly assembly underpinning built-in asymmetry in the yeast centrosome duplication cycle requires cyclin-dependent kinase | ftp://odr.stowers.org/LIBPB-1526 | Stowers Original Data Repository, LIBPB-1526 |

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
