## [Decision Letter]

**Acceptance summary:**

Asymmetric inheritance of the centrosome occurs in self-renewing stem cell divisions and is also observed in the yeast centrosome equivalent, the spindle pole body. Using compelling super resolution imaging, the study identifies CDK-dependent phosphorylation as determinant of the preferential recruitment of microtubule nucleation activity to the old spindle pole body, promoting asymmetric formation of astral microtubules and segregation in the bud. The data supports the view that apart from cortical cues this inheritance pattern also depends on a spindle pole body-intrinsic asymmetry.

**Decision letter after peer review:**

Thank you for submitting your article "Orderly assembly underpinning asymmetry in the yeast centrosome duplication cycle requires cyclin-dependent kinase" for consideration by *eLife*. Your article has been reviewed by three peer reviewers, and the evaluation has been overseen by a Reviewing Editor and Anna Akhmanova as the Senior Editor. The reviewers have opted to remain anonymous.

The reviewers have discussed the reviews with one another and the Reviewing Editor has drafted this decision to help you prepare a revised submission.

Summary:

Asymmetric inheritance of the centrosome has been linked to self-renewing stem cell divisions. The yeast spindle pole body (SPB), a centrosome-equivalent structure, is also inherited non-randomly, with the old SPB preferentially segregating to the new daughter cell. Different types of mechanisms that control this asymmetry have been proposed. In this study the authors corroborate that there is an inherent structural asymmetry based on Spc72 localization to the old SPB during the duplication stage and that the new SPB does not acquire Spc72 and nucleation complexes until later during mitotic spindle assembly. They show that the asymmetry is controlled by CDK-dependent phosphorylation of Nud1, which increases affinity for the nucleation complex receptor Spc72 and thus its preferential retainment at the old SPB during duplication. In addition, CDK is shown to control the ordered assembly of the new SPB, including the correct distribution of nucleation complexes to the cytoplasmic and nuclear faces of the SPB.

The reviewers were impressed with the beautiful SIM analysis supporting the SPB age/maturation hypothesis as mechanism underlying asymmetric inheritance. However, there was some concern regarding the phosphorylation-based mechanism as basis for asymmetry. The following issues should be addressed:

Main points:

1) It is clear that CDK plays an important role in SPB asymmetry and that Nud1 is a CDK substrate. However, whether Nud1 phosphorylation by CDK is critical for SPB asymmetry is less convincing. Considering that the impact of non-phosphorylatable *Nud1^7A^* is relatively subtle (Figures 4D and 5F) the importance of Nud1 phosphorylation for asymmetry should be discussed without overstating it. The authors should also discuss the study by Lengefeld et al., 2018, and the differing view in this context in more detail. Apart from referring to differences in methodology, can the current results be integrated with the role of Kar9 and spatial cues?

2) Text/Figure improvements

Figures 1 D-F

a) Figure 1D and the top row of Figure 1E both show that Spc72 is recruited to the new pole over time, and eventually reaches a symmetric distribution at both poles. Please provide the criteria distinguishing all the categories (especially strongly asymmetric from asymmetric).

b) The legends of the graphs in Figure 1D contain "(OP)" for the Tub4 line scans but the Spc72 line scans do not. Please clarify the meaning. Showing on the figure where the linescan was drawn would clarify what was measured (perhaps in a separate panel).

c) Figure 1F seems to be shown in order to prove that the kar1∆15 mutation does not affect spindle length, but this is not mentioned in the text.

According to the manuscript text, Figure 2B c-e shows that in KAR1 cells, the side-by-side stage is denoted "by the presence of duplicated Spc42 signals with two Tub4 inner plaque foci". However, Figure 2B c and d show a single Tub4 inner plaque focus. “…outer plaque (Figure 2B, b, big and small arrowheads, respectively). After appearance of the new inner plaque (Figure 2B, c-e), label continued to distribute between those two locations” more accurately describe the representative cells, and so we suggest a small revision to the first sentence to more accurately describe the heterogeneity.

The Discussion is not clearly written and would benefit from editing and shortening. A few examples: What does "this" refer to “ between *cdc28-4* (a G1 hypomorph *CDK* allele) and *clb5Δ* promoting this phenotype”? In paragraph two should a reference be included for the paper that used dSTORM? “ Our use of a second label and three-dimensional realignment…” begins a run-on sentence that ends seven lines later and includes several important points. “…when CDK sites are cancelled in multiple components (Jones et al., 2018).”: does cancelled mean mutated? Sentence starting with "The extent" is unclear. The thesis of the paragraph starting in paragraph six is difficult to discern.

A scheme at the end would help the reader to gather the main conclusions.

---

## [Author Response]

Main points:1) It is clear that CDK plays an important role in SPB asymmetry and that Nud1 is a CDK substrate. However, whether Nud1 phosphorylation by CDK is critical for SPB asymmetry is less convincing. Considering that the impact of non-phosphorylatable Nud1^7A^ is relatively subtle (Figures 4D and 5F) the importance of Nud1 phosphorylation for asymmetry should be discussed without overstating it. The authors should also discuss the study by Lengefeld et al., 2018, and the differing view in this context in more detail. Apart from referring to differences in methodology, can the current results be integrated with the role of Kar9 and spatial cues?

We agree with the reviewers that Spc72 asymmetry was not as affected by *nud1^7A^* compared to *cdc28-4 clb5∆* (~20% vs.~50%, respectively relative to wild type). Therefore, we state in the manuscript that *nud1^7A^* did not fully recapitulate the loss of S-phase CDK thus implicating other substrates at the SPB or beyond (including a number of protein kinases known to phosphorylate Nud1 that require CDK for their activity).

However, what we found most unexpected and surprising was that this ~20% decrease in Spc72 asymmetry seen in unseparated SPBs in the *nud1^7A^* mutant translated in substantial symmetry during spindle assembly, with both Spc72 and γTC detected at the new SPB in 100% of cells (Figure 5A-E). The increase in nucleation sites at the new SPB during spindle assembly is consistent with the perturbation to the program for establishment of spindle polarity and the modest but reproducible impact on the pattern of SPB inheritance. This finding highlights the central role that Nud1 plays in the control of spindle polarity by its involvement in astral microtubule (aMT) organisation via Spc72.

Our data both extends and contrasts to previous work in several ways. First, previous work from the Barral lab showed that a *nud1^ts^* allele also randomised SPB inheritance pointing to the importance of Nud1 for this process. Yet their study linked instead Nud1 function to Kar9 (via Dbf2 activity at metaphase), suggesting that Nud1 functioned not through aMTs but through the Mitotic Exit Network (MEN). Later, the Amon lab showed that MEN activation is confined to a post-anaphase interval and does not control Kar9 polarity (Campbell et al., 2019). Thus, it was essential for us to revisit the role of Nud1 control of spindle polarity by its involvement in aMT organisation via Spc72 as part of our study of the intrinsic or built-in asymmetry.

The idea that spatial cues are integral to the program of spindle polarity in interplay with intrinsic SPB asymmetry is an important concept that has developed over several decades of research. Early studies from the Bloom lab showed that aMT asymmetry underlies spindle polarity —the old SPB that initially carries aMTs is destined to the bud (Shaw et al., 1997). Moreover, upon loss of Bud6 or Kar9 function through knockout, bud-ward fate is no longer linked to the pole first containing aMTs (Yeh et al., 2000) without otherwise preventing aMT asymmetry. It follows that age-dependent aMT asymmetry can only translate into differential SPB fate in the hands of cortical determinants, underscoring the instructive importance of spatial cues (for review see Geymonat and Segal, 2017). In other words, aMTs at the old SPB cannot dictate bud-ward targeting without the aid of spatial cues from the cell cortex. Both *bud6∆* and *kar9∆* mutants randomise the pattern of SPB inheritance, irrespective of intrinsic SPB identity, without negating the importance of the intrinsic pathway.

The Lengefeld et al. study claims that spatial cues are the sole contributors to instigate aMT asymmetry. While their observations are consistent with a large body of work showing that cortical cues are important for aMT asymmetry (see Juanes et al., 2013,; Cepeda-Garcia et al., 2010; Ten Hoopen et al., 2012; Huisman et al., 2004), their inference overlooks the established principle that cortical cues are necessary but not sufficient to link SPB age, identity and fate.

While we do not dispute the notion that intrinsic SPB asymmetry cannot translate into asymmetric fate without extrinsic polarity cues, we show the mechanistic basis for SPB and aMT asymmetry in this paper. Using the same genetic background as Lengefeld et al., we observe Spc72 asymmetry (Figure 1—figure supplement 5). Our ability to spatially resolve and visualize the outer plaque during SPB duplication/maturation in unperturbed cycling, wild-type cells distinguishes our work from that of Lengefeld et al. Further, we observe Spc72 asymmetry in four unrelated yeast genetic backgrounds (Figure 1—figure supplement 5). Taken together, our work strongly suggests that both Spc72 and γTC asymmetry is a general premise in the SPB duplication cycle.

To address the reviewers' comment we have edited the manuscript in order that the role of direct CDK phosphorylation of Nud1 not be overstated, in line with the evidence presented. The Discussion has been edited to better place our findings in the broader context including consideration of Lengefeld et al., 2018. Here we provide a brief overview on the role of cortical cues and their implications to the findings reported to fit within the space of the Discussion.

2) Text/Figure improvementsFigures 1D-Fa) Figure 1D and the top row of Figure 1E both show that Spc72 is recruited to the new pole over time, and eventually reaches a symmetric distribution at both poles. Please provide the criteria distinguishing all the categories (especially strongly asymmetric from asymmetric).

Spindles were arbitrarily classified according to the difference in label intensity between the SPBs as follows: "one pole", label visible at one outer plaque only; "strongly asymmetric", at least an 8-fold difference in label intensity between SPBs; "asymmetric", between 8-fold and 1.3-fold difference; "symmetric", less than 1.3-fold difference. This information has been now included under Materials and methods.

b) The legends of the graphs in Figure 1D contain "(OP)" for the Tub4 line scans but the Spc72 line scans do not. Please clarify the meaning. Showing on the figure where the linescan was drawn would clarify what was measured (perhaps in a separate panel).

In all cases in Figure 1D, a single 3-px width linescan sampled SPB labels along the respective OP-IP axis. In the Tub4 channel this results in a typical spindle profile with two flanking weaker peaks (outer plaques) and two internal stronger peaks (inner plaques). Classification of Tub4 refers to the relative outer plaque label only. We have clarified this in the figure legend and have annotated a linescan to point at the IP and OP peaks.

c) Figure 1F seems to be shown in order to prove that the kar1∆15 mutation does not affect spindle length, but this is not mentioned in the text.

Yes, kar1∆15 does not affect spindle length. As a result, wild type and mutant samples obtained from asynchronous cell populations did not differ with regard to the distribution of spindles by length. Furthermore, profiles are shown to establish that any differences regarding asymmetric marking do not arise from any possible differences in length distribution. We have amended the text to include a reference to Figure 1F.

According to the manuscript text, Figure 2B c-e shows that in KAR1 cells, the side-by-side stage is denoted "by the presence of duplicated Spc42 signals with two Tub4 inner plaque foci". However, Figure 2B c and d show a single Tub4 inner plaque focus. “…outer plaque (Figure 2B, b, big and small arrowheads, respectively). After appearance of the new inner plaque (Figure 2B, c-e), label continued to distribute between those two locations” more accurately describe the representative cells, and so we suggest a small revision to the first sentence to more accurately describe the heterogeneity.

Figure 2B c-d present typical morphologies of the side-by-side stage with two Tub4 inner plaque foci. Following the reviewers' recommendation, we have edited the text for clarity and added a different set of arrows to point at inner plaques.

The Discussion is not clearly written and would benefit from editing and shortening. A few examples: What does "this" refer to “ between cdc28-4 (a G1 hypomorph CDK allele) and clb5Δ promoting this phenotype”? In paragraph two should a reference be included for the paper that used dSTORM? “ Our use of a second label and three-dimensional realignment…” begins a run-on sentence that ends seven lines later and includes several important points. “…when CDK sites are cancelled in multiple components (Jones et al., 2018).”: does cancelled mean mutated? Sentence starting with "The extent" is unclear. The thesis of the paragraph starting in paragraph six is difficult to discern.A scheme at the end would help the reader to gather the main conclusions.

We have edited the Discussion for length but elaborated on Lengefeld et al. as requested above. We have also included a cartoon (Figure 7) summarising our key conclusions.